# GFI1 tethers the NuRD complex to open and transcriptionally active chromatin in myeloid progenitors

Anne Helness[1], Jennifer Fraszczak[1], Charles Joly-Beauparlant[2], Halil Bagci[1,7], Christian Trahan[1], Kaifee Arman [1], Peiman Shooshtarizadeh[1], Riyan Chen[1], Marina Ayoub[1,8], Jean-François Côté [1,3,4,5], Marlene Oeffinger[1,4], Arnaud Droit [2] & Tarik Möröy [1,5,6 ✉]

Growth factor independent 1 (GFI1) is a SNAG-domain, DNA binding transcriptional repressor which controls myeloid differentiation through molecular mechanisms and co-factors that still remain to be clearly identified. Here we show that GFI1 associates with the chromodomain helicase DNA binding protein 4 (CHD4) and other components of the Nucleosome remodeling and deacetylase (NuRD) complex. In granulo-monocytic precursors, GFI1, CHD4 or GFI1/CHD4 complexes occupy sites enriched for histone marks associated with active transcription suggesting that GFI1 recruits the NuRD complex to target genes regulated by active or bivalent promoters and enhancers. GFI1 and GFI1/CHD4 complexes occupy promoters that are either enriched for IRF1 or SPI1 consensus binding sites, respectively. During neutrophil differentiation, chromatin closure and depletion of H3K4me2 occurs at different degrees depending on whether GFI1, CHD4 or both are present, indicating that GFI1 is more efficient in depleting of H3K4me2 and -me1 marks when associated with CHD4. Our data suggest that GFI1/CHD4 complexes regulate histone modifications differentially to enable regulation of target genes affecting immune response, nucleosome organization or cellular metabolic processes and that both the target gene specificity and the activity of GFI1 during myeloid differentiation depends on the presence of chromatin remodeling complexes.

---

[1] Institut de recherches cliniques de Montréal, Montréal, QC H2W 1R7, Canada. [2] Département de Médecine Moléculaire, Faculté de Médecine, Université Laval, Québec, QC, Canada. [3] Department of Anatomy and Cell Biology, McGill University, Montréal, QC H3A 0C7, Canada. [4] Département de Biochimie, Université de Montréal, Montréal, QC H3C 3J7, Canada. [5] Division of Experimental Medicine, McGill University, Montreal, QC, Canada. [6] Département de Microbiologie, Infectiologie et Immunologie, Université de Montréal, Montréal, QC, Canada. [7] Present address: Institute for Biochemistry, ETH Zürich, Zürich, Switzerland. [8] Present address: Hôpital pour Enfants, Ste Justine, Montreal, QC, Canada. ✉email: Tarik.Moroy@ircm.qc.ca

The DNA-binding zinc finger proteins GFI1 and GFI1B act as transcriptional repressors by recruiting the complex containing the lysine-specific histone demethylase 1A (LSD1, official name KDM1A) and its cofactor CoREST (official name RCOR1) together with HDACs to sites of specific target genes that harbor a GFI/GFI1B consensus DNA binding motif[1] (for a review see refs. [2–5]). GFI1 is critical for the differentiation of myeloid cells into neutrophils, which is highlighted for example by GFI1 deficient mice that entirely lack neutrophil granulocytes and, as a consequence, show major defects in their innate immune response[6,7]. GFI1 and its shorter paralog GFI1B share a 20 aa N-terminal SNAG domain that shows sequence similarity to the N-terminal tail of histone H3[8]. It has been suggested that the GFI1/B SNAG domains and the H3 N-terminus can compete for binding to the same pocket in the LSD1 protein[8]. Transcriptional repression is achieved by the enzymatic action of LSD1 and HDACs leading to the demethylation of histone H3 Lysine 4 (H3K4) and the deacetylation of histone H3 Lysine 9 (H3K9)[1,9–11]. The general applicability of this model has been challenged by recent observations indicating that H3K4 methylation states do not change in cells upon treatment with an LSD1 inhibitor that not only blocks its enzymatic activity but also leads to the eviction of GFI1 and LSD1 from promoter sites[12,13]. Moreover, this report finds that LSD1's demethylase function is not critical for GFI1 function but rather suggests that LSD1's physical interaction with GFI1's SNAG-domain is crucial and that LSD1 rather serves as a scaffold for other histone-modifying enzymes such as HDACs[12,13]. However, further investigation on this matter is necessary since this study was based on a presumed inactive LSD1 mutant for which another study showed some catalytic activity[14].

To understand the precise molecular function of GFI1 as a transcriptional regulator, it is necessary to identify the epigenetic modifier complexes that are recruited by this GFI1/LSD1 scaffold and act on chromatin structure and more specifically on histone modifications. Indeed, it was recently shown that GFI1B can recruit members of the so-called BRAF-histone deacetylase (HDAC) (BHC) chromatin–remodeling complex that contains LSD1, members of the CoREST complex (RCOR1, -2 and -3), HDACs as well as a number of the high mobility group of proteins (HMG20A and -B) and other associated proteins[15]. The nucleosome remodeling and deacetylase (NuRD) complex would be an excellent candidate as well, since similar to the BHC complex it also facilitates histone deacetylase mediated chromatin condensation and some studies have reported that it also contains LSD1[16–18].

The NuRD complex differs however from the BHC complex in that it contains seven different proteins divided into two sub-complexes, one which comprises an ATP-dependent nucleosome remodeling activity and another which harbors HDACs targeting H3K9 or histone H3 Lysine 27 (H3K27)[19–21]. Characteristic for the NuRD complex are its major components, the closely related proteins CHD4 (chromodomain helicase DNA binding protein 4 or Mi-2β), CHD3, and CHD5[22]. CHD4 contains an SNF helicase domain and PhD/Chromo domains that mediate its interaction between nucleosomes and methylated histones[23]. The methyl-CpG-binding domain proteins MBD2 and MBD3 represent its non-enzymatic components and link the ATPase remodeling activities to the HDAC1 and -2 containing subcomplexes. The metastasis-associated proteins MTA1, MTA2, and MTA3 are also part of this subcomplex and mediate binding to DNA, to HDAC1, and to other transcription factors that can recruit NuRD to specific loci in the genome[24–26]. The proteins RBBP7 and RBBP4 bind histones and most likely have roles as scaffolds[27,28]. The GATA zinc-finger domain-containing proteins GATAD2A and -B interact with MBD2/3 and are also canonical members of the NuRD complex, although their precise function remains to be determined[29].

The NuRD complex can mediate both transcriptional repression or activation[16,30] and can be recruited to sites of bivalent or poised targets in chromatin that are primed to be efficiently activated or repressed by modifying the histone marks during progenitor self-renewal or differentiation[31,32]. Bivalent target promoters or genomic loci with enhancers show both repression marks such as H3K27 trimethylation (H3K27me3) and activation marks such as H3K4 dimethylation (H3K4me2) or H3K27 acetylation (H3K27ac) at the same time and also feature modifications such as H3K4 monomethylation (H3K4me1), which identity so-called primed or induced enhancers[31]. To exert its function in a tissue and differentiation stage-specific manner, NuRD associates for instance during lymphoid development with lineage-specific transcription factors and co-regulators[30], such as IKZF1 (IKAROS), BCL6, or BLIMP1[33–35].

Here, we show that GFI1 and GFI1B interact with members of the NuRD complex, notably with the chromodomain helicase DNA binding protein 4 (CHD4). In myeloid progenitors, GFI1 occupies chromatin together with CHD4 at specific target regions that are different from those regions occupied by GFI1 or CHD4 alone. These target regions bear characteristics of open chromatin and histone modifications that are associated with active transcription or poised enhancers. GFI1 occupies promoters and genomic sites that are different from those occupied by GFI1/CHD4 complexes and are upstream of different groups of genes that can be distinguished by the enrichment of either IRF1 or SPI1 binding consensus sequences, respectively. During neutrophil differentiation, different levels of chromatin closure and a reduction of H3K4me levels are seen depending on whether sites are occupied by GFI1, CHD4, or GFI1/CHD4 complexes. Lastly, GFI1, CHD4, or GFI1/CHD4 occupy promoters of genes that fall into three distinct groups termed immune system, chromatin/nucleosome assembly, and metabolic process, respectively, that can be both upregulate and downregulate during neutrophil differentiation.

## Results

**GFI1 associates with the nucleosome–remodeling, and histone deacetylase (NuRD) complex.** We had used AP–MS (affinity purification and mass spectrometry) to identify proteins that co-purified with Flag-tagged versions of GFI1 and GFI1B in HEK293T cells[36,37]. This approach revealed the presence of members of the NuRD complex such as chromodomain helicase DNA binding proteins 3 and -4 (CHD3 and -4), metastasis-associated 1, and -2 (MTA1 and -2), and the chromatin remodeling factor retinoblastoma binding protein 4 (RBBP4, also called chromatin assembly factor 1 or CAF-1) in both isolated GFI1 and GFI1B complexes (Fig. 1A, Suppl. Fig. 1A). The peptide coverage for these factors was in a similar range as for those proteins that are known to associate with GFI1 and GFI1B such as HDAC1 and members of the CoREST complex such as RCOR1, 2, or -3 (Fig. 1A). To validate these findings, we used a BioID approach in HEK293T cells for GFI1 associated proteins, compiled the data with known interactions from the IntAct and BioGrid databases[38,39], and observed that GFI1 has the potential to associate with four major complexes, notably with the NuRD complex, in agreement with our findings from the AP–MS experiment, but also with the SWI/SNF, CtBP, and cohesin complexes (Fig. 1B, Suppl. Data 1).

Next, we compared the data with our previously reported BioID experiment for GFI1B[37] and found again members of the NuRD complex as high ranking candidates for binding partners of both GFI1 and GFI1B (Fig. 1B, Suppl. Table 1), and proteins

**a**

| Gene Symbol | GFI1–FLAG | | | GFI1B–FLAG | | | Reference | Accession | Complex |
|---|---|---|---|---|---|---|---|---|---|
| | Unique Peptides | Total Peptides | Coverage | Unique Peptides2 | Total Peptides2 | Coverage4 | | | |
| CHD4 (Mi–2β) | 28 | 76 | 17% | 58 | 92 | 33% | CHD4_HUMAN | gi1107696 | NuRD |
| MTA2 | 3 | 4 | 6% | 24 | 37 | 31% | MTA2_HUMAN | gi14141170 | NuRD |
| RBBP4 | nd | nd | nd | 10 | 15 | 21% | RBBP4_HUMAN | gi207029415 | NuRD/CoRest |
| HDAC1 ✱ | 6 | 12 | 31% | 29 | 45 | 45% | HDAC1_HUMAN | gi13128860 | NuRD/CoRest |
| HDAC2 ✱ | 24 | 51 | 49% | 17 | 29 | 59% | HDAC2_HUMAN | gi119568637 | NuRD/CoRest |
| LSD1 ✱ | nd | nd | nd | nd | nd | nd | LSD1_HUMAN | gi3043726 | NuRD/CoRest |
| RCOR1 ✱ | 21 | 44 | 45% | 26 | 43 | 44% | RCOR1_HUMAN | gi119602199 | CoREST |
| RCOR2 | 2 | 4 | 7% | 2 | 3 | 6,40% | RCOR2_HUMAN | gi119594587 | CoREST |
| RCOR3 | 9 | 29 | 20% | 16 | 24 | 47% | RCOR3_HUMAN | gi8922733 | CoREST |
| GFI1 | 41 | 196 | 75% | nd | nd | nd | GFI1_HUMAN | gi71037377 | |
| GFI1B | nd | nd | nd | 32 | 85 | 71% | GFI1B_HUMAN | gi205277343 | |

**b**

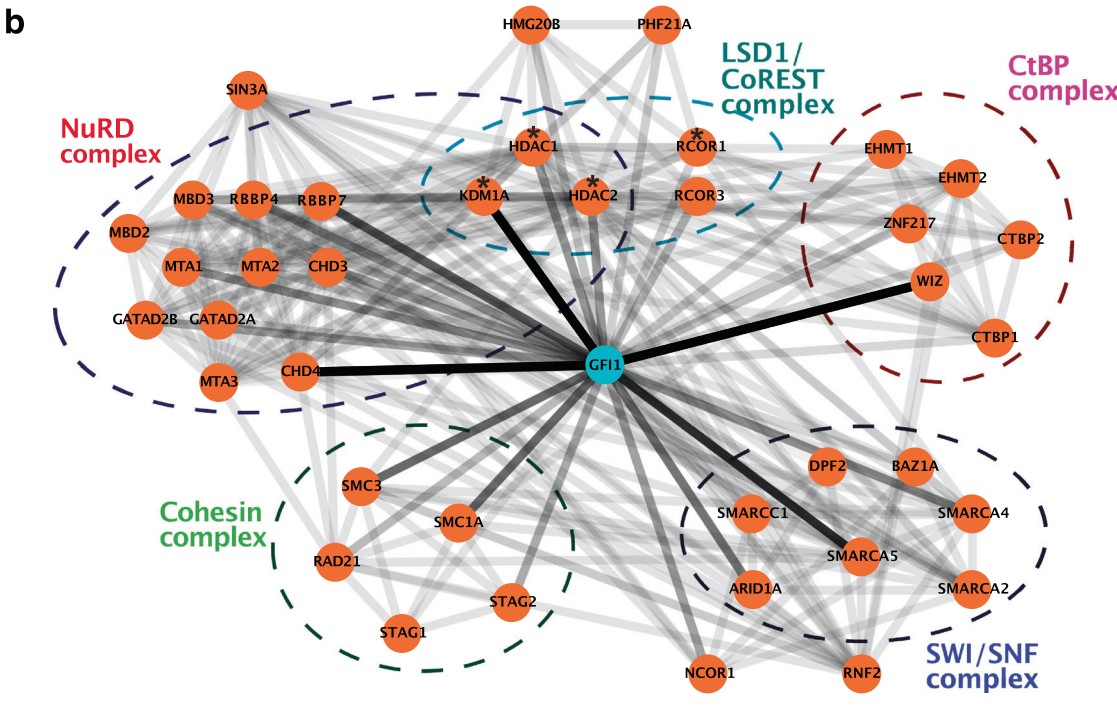

**c**

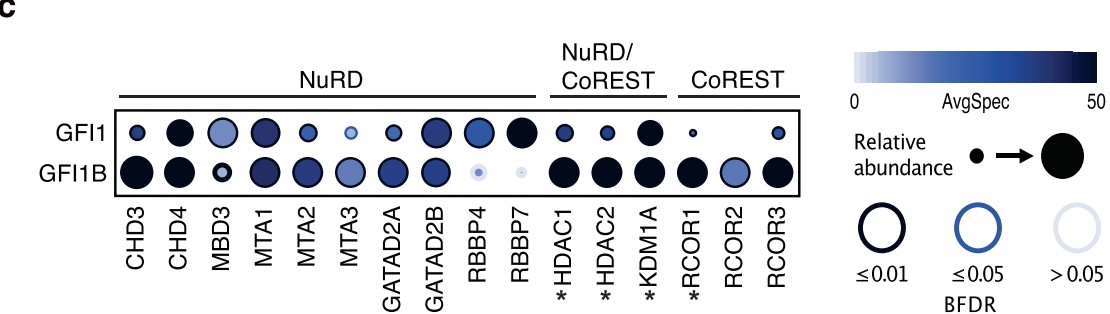

already known to bind to these two factors such as HDAC1 and -2, LSD1 and the CoREST proteins RCOR1, -2, and -3 (Fig. 1C, Suppl. Data 1). Quantification of the mass spectrometry results showed high peptide coverage of CHD3 and CHD4, MTA1, -2, -3, and GATA Zinc Finger Domain Containing 2A and 2B (GATAD2A and GATAD2B); the abundance of recovered peptides for GFI1B being higher than for GFI1 (Fig. 1C). Also

detectable were peptides for other NuRD complex members such as the methyl-CpG binding domain protein 3 (MBD3) and RBBP4 and -7 (Fig. 1C, Suppl. Data 1).

**NuRD components CHD4 and MTA2 associate with GFI1 C- and N-termini.** Next, we expressed Flag-tagged versions of GFI1 and GFI1B in HEK293T cells, precipitated NEs with anti-Flag

**Fig. 1 GFI1 is associated with protein components of the NuRD complex. a** Result from AP-MS (affinity purification and mass spectrometry) experiments with GFI1- or GFI1B-Flag-tagged fusion proteins expressed in HEK293T cells for NuRD and LSD1/CoRESTcomplexes. *: previously known GFI1 interacting proteins. **b** Interaction map of the GFI1 proteome obtained through BioID data. The interactions between GFI1 and the preys are "weighted" in color according to AvgSpec (e.g., GFI1-KDM1A: 104 AvgSpec, darker color; GFI1-HDAC2: 44 AvgSpec less dark color). The known Prey–Prey interactions (e.g., KDM1A-HDAC1) were imported from databases IntAct and BioGrid, and are represented by fewer dark connections since they cannot be normalized to the GFI1-BioID screen. GFI1–prey interactions remain[31] visible. *: previously known GFI1 interacting proteins. **c** Dot Plot showing BioID interactions of GFI1-BirA*-Flag or GFI1B-BirA*-Flag with the indicated members of the NuRD or LSD1/CoREST complexes. The node color depicts the average spectral counts. The relative abundance of prey vs. the bait is shown by the circle size. The edge color represents the confidence score of the BioID/MS interaction (5% < BFDR as low confidence score, 1% < BFDR ≤ 5% as medium confidence or BFDR ≤ 1% as high confidence). *BFDR* Bayesian false discovery rate. *: previously known GFI1-interacting proteins.

agarose, and analyzed the collected proteins per western blot. We were able to detect CHD4, MTA2, RBBP4/6, and as positive controls, HDAC1 and LSD1 with both GFI1 and GFI1B (Fig. 2A, Suppl. Fig. 8). Also, samples from THP-1 cell extracts incubated with an anti GFI1 antibody contained both CHD4 and MTA2 protein (Fig. 2B). Conversely, an anti-MTA antibody precipitated both GFI1 and CHD4 in THP-1 cells (Fig. 2B) demonstrating that both CHD4 and MTA2 can associate with GFI1 at endogenous expression levels (Fig. 2B, Suppl. Fig. 8). In addition, Flag-tagged GFI1 and MTA2 showed similar interactions in the presence or absence of Ethidium-bromide or Benzonase (Suppl. Fig. 1B, Suppl. Fig. 9), indicating that this interaction is independent of the presence of DNA.

To determine the region of the association between GFI1 and CHD4 or MTA2, we transfected different Flag-tagged truncation- and deletion mutants of GFI1 into HEK293T cells and precipitated extracts with anti-CHD4 or anti MTA2 antibodies. Analysis of the collected proteins by Western blot revealed that deletion of the SNAG domain weakened the interaction with either CHD4 or MTA2 and that the presence of the SNAG- and the zinc finger domains alone could maintain interaction with these two proteins (Fig. 2C, Suppl. Fig. 8). Neither CHD4 nor MTA2 antibodies did immunoprecipitated the truncated GFI1 protein N152–258-Flag (Fig. 2C, Suppl. Fig. 8), suggesting that this region of GFI1 does not participate in the association with NuRD complex members. A BioID experiment with either the full-length GFI1 or a truncated version GFI1 lacking the 20 aa N-terminal SNAG domain (GFI1ΔSNAG) as baits confirmed that the interaction of GFI1 with the LSD1/CoREST complex requires the SNAG domain (Fig. 2D, Suppl. Fig. 1C, Suppl. Data 1). The experiment also showed that the interaction of GFI1 with members of the NuRD complex such as CHD3, −4, MBD3, MTA3, GATAD2A, and HDAC1 and -2 seemed to be dependent on the presence of the SNAG domain. In contrast, an association of RBBP7 and/or GATAD2B with GFI1 is still detectable with the ΔSNAG mutant indicating that they could associate with another region of GFI1 such as the ZF domain (Fig. 2D). A GO term analysis indicated that biological processes including nucleosome disassembly and protein acetylation and the molecular functions transcription coregulator and lysine-acetylated histone binding were lost in the BioID with GFI1ΔSNAG (Suppl. Fig. 1D, E, Suppl. Data 2).

Nuclear GFI1 from Kasumi cell extracts was eluted by size exclusion chromatography (SEC) with peaks at around 2 MDa and 0.5 MDa (Fig. 2E, Suppl. Fig. 8). While the GFI1 2 MDa complex did not contain substantial amounts of LSD1, RCOR1, MTA2, or CHD4, all four proteins were found in the complex with GFI1 eluting at around 0.5 MDa (Fig. 2E, Suppl. Fig. 8). Immunoprecipitation of extracts from transfected HEK293T cells expressing a Flag-tagged version of GFI1 confirmed that LSD1, RCOR1, and MTA2 can associate with GFI1 (Suppl. Fig. 1F, Suppl. Fig. 10). Both GFI1 and LSD1 but not RCOR1 were enriched in extracts precipitated anti MTA2 antibodies from

HEK293T cells expressing a Flag-tagged version of GFI1 (Fig. 2F, Suppl. Fig. 8). In the absence of GFI1, anti MTA2 antibodies could still precipitate LSD1, albeit at lower levels (Fig. 2F, Suppl. Fig. 8). Anti-RCOR1 antibodies precipitated both GFI1 and LSD1 but to a much lesser extent MTA2 regardless of whether GFI1 was expressed or not (Fig. 2F, Suppl. Fig. 8). This suggests that GFI1 may interact with components of the NuRD complex such as MTA2 independently from the LSD1/CoREST complex.

**GFI1 and CHD4 co-occupy sites of open chromatin and active transcription.** We chose primary murine GMPs (lin⁻kIT⁺SCA⁻CD16/32⁺CD34⁺ cells FACS-sorted from bone marrow, Suppl. Fig. 11), in which GFI1 is expressed[40] to further investigate the association between NuRD and GFI1. Chromatin Immunoprecipitation and sequencing (ChIP-seq) experiments with antibodies against murine GFI1 and CHD4 in wild type (WT) GMPs showed 3188 peaks for GFI1 and 4236 peaks for CHD4 and revealed that at 1128 sites occupation of GFI1 and CHD4 overlapped (Fig. 3A). The sites that are co-occupied by GFI1 and CHD4 are primarily at promoter- (~30%) and intergenic regions (~38%) (Fig. 3B). When we compared the overall binding of CHD4 to chromatin in sorted GMPs from WT with gene-deficient (*Gfi1⁻/⁻*) mice, we observed that of the 1128 sites that were co-occupied by both GFI1 and CHD4 in WT cells, 841 lost enrichment of CHD4 when GFI1 was absent (Fig. 3C, D), most of them in regions <3 kb from the TSS. At the remaining 287 sites, CHD4 was still enriched regardless of GFI1's presence or absence (Fig. 3C, D). Except for 9 CHD4 enriched sites, no significant gain in CHD4 bound sites was observed in *Gfi1⁻/⁻* GMPs (Fig. 3C). The 3108 sites that were occupied only by CHD4 in WT cells, remained largely unaffected by *Gfi1* deficiency, i.e., CHD4 remained at these sites (Fig. 3C).

Examples for loci occupied by GFI1/CHD4 are the genes *Cd34*, *Csf1r* (encoding M-CSFR), or *Csf1* (encoding M-CSF) (Fig. 3E, Suppl. Fig. 2A). *Gfi1* deletion is associated with a loss of CHD4 at these sites and RNA-seq data showed that this leads to the upregulation of *Cd34* mRNA (Fig. 3E) but does not affect *Csf1r* and *Csf1* mRNA levels (Fig. 3F and Suppl. Fig. 2A, B, Suppl. Table 3). For further validation, we chose five other genes selected from the group of 841 loci where CHD4 binding was lost in the absence of GFI1 according to the ChIP-seq data. ChIP-qPCR on these gene loci for CHD4 in primary WT and *Gfi1⁻/⁻* GMPs showed loss of CHD4 enrichment confirming the results of the CHD4 ChIP-seq experiment (Suppl. Fig. 2C, Suppl. Table 3). RNA-seq and flow cytometric analysis demonstrated that *Chd4* mRNA and CHD4 protein levels were not altered in *Gfi1⁻/⁻* GMPs compared to WT GMPs (Suppl. Fig. 2D, E). Genes that lost CHD4 binding at their promoters in the absence of GFI1 or maintained CHD4 did not show a significantly different upregulation or downregulation of gene expression in *Gfi1⁻/⁻* vs. wt GMPs (Fig. 3G). Also, upregulation or downregulation of GFI1 target gens in *Gfi1⁻/⁻* vs. wt GMPs did not correlate with the presence or absence of CHD4 (Suppl. Fig. 3A, B, Suppl.

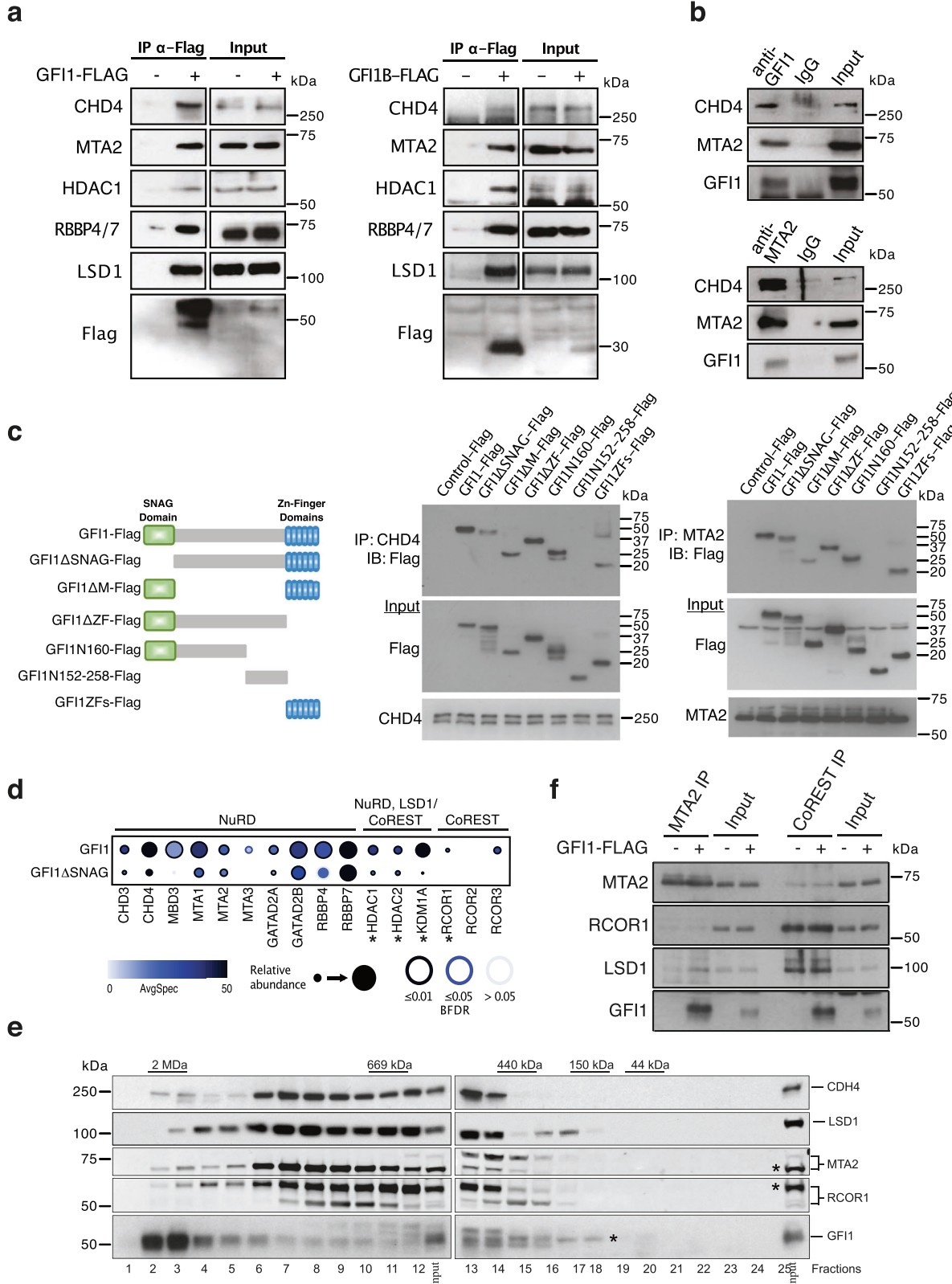

Table 3). This suggested that the deletion of GFI1 affects gene expression independently of the presence of CHD4.

To test whether GFI1/CHD4 complexes affect chromatin remodeling, we performed ATAC-seq and ChIP-seq analyses of GMPs with antibodies against methylated and acetylated histone H3. The obtained data were filtered for promoter regions defined here as regions located between less than 2 kb upstream and less than 500 bp downstream of the transcription start sites (TSS), and separately also for enhancer regions selected based on the criteria defined in the Fantom5 enhancer atlas[41]. The data were ordered according to CHD4 occupation and were separated into three groups: occupation by CHD4 alone, by GFI1/CHD4 together, or by GFI1 alone (Fig. 4A). We observed that promoters occupied by GFI1 show higher levels of markers associated with active

**Fig. 2 Association of GFI1 with NuRD complex requires the SNAG domain. a** GFI1-Flag fusion protein was immunoprecipitated from HEK293T cells; precipitates were separated by SDS-PAGE and blotted for the indicated proteins. Lanes with extracts from cells transfected or not with constructs for the expression of GFI1-Flag fusion proteins are labeled as + or −, respectively. **b** Immunoprecipitation with anti GFI1 or anti MTA2 antibodies from THP-1 cell extracts. Precipitates were analyzed by Western blot for the presence of CHD4, MTA2, or GFI1. **c** Schema of different GFI1-Flag fusion proteins that were expressed in HEK293T cells. Extracts of transfected cells were precipitated with anti CHD4 or MTA2 antibodies, separated by SDS-PAGE, and analyzed by Western blotting with an anti-Flag antibody. **d** Dot Plot showing BioID interactions of GFI1-BirA*-Flag or GFI1 lacking the SNAG domain (GFIΔSNAG-BirA*-Flag) with the indicated members of the NuRD or LSD1/CoREST complexes. The node color depicts the average spectral counts. The relative abundance of prey versus the bait is shown by the circle size. The edge color represents the confidence score of the BioID/MS interaction (5% < BFDR as low confidence score, 1% < BFDR ≤ 5% as medium confidence or BFDR ≤ 1% as high confidence). BFDR Bayesian false discovery rate. *: previously known GFI1 interacting proteins. **e** Nuclear extracts from Kasumi 1 cells were fractionated using a Superose 6 10/300GL column; 0.5 ml fractions were collected, TCA-precipitated and loaded on SDS-PAGE for western blot analysis; immunoblots were probed with the indicated antibodies. Asterisks indicate the bands representing the MTA2 and RCOR1 proteins that associate with GFI1, as shown in the immunoprecipitation experiment in Suppl. Fig. 1F. The asterisk next to GFI1 indicates the bands that represent the GFI1 protein according to the input. It is likely that the bands represent differentially modified forms of the GFI1 protein since they are recognized by the anti GFI1 antibody. **f** HEK293T cells were transfected to express a full-length Flag-tagged GFI1 protein. Extracts were immunoprecipitated with anti-MTA2 or anti-RCOR1 antibodies and the precipitates were analyzed by Western blot for the presence of MTA2, RCOR1, LSD1, or GFI1. Lanes with extracts from cells transfected or not with constructs for the expression of GFI1-Flag fusion proteins are labeled as "+" or "−", respectively.

transcription such as H3K4me3 and H3K27ac than the promoters occupied by CHD4 alone. Promoters co-occupied by both GFI1 and CHD4 showed intermediate levels of H3K4me3 and H3K27ac (Fig. 4B). In contrast, promoters occupied by CHD4 showed higher levels of H3K27me3 H3K9me3 and H3K4me2, all associated with transcriptional repression, than promoters occupied by GFI1 or by both GFI1 and CHD4 (Fig. 4B), with a similar pattern for the levels of H3K4me2 and H3K4me3, which are markers of transcriptional activation (Fig. 4B). This situation is exemplified by the promoter regions of the genes encoding the GFI1 targets *Csf1* (Fig. 4C) and *Csf1r* (suppl. Fig. 4), which are both expressed in GMPs (Suppl. Fig. 2A, B, Suppl. Table 3 and Fig. 3F).

H3K9 and H3K27 methylation and acetylation patterns at enhancers correlated similarly with GFI1 and CHD4 occupation to those at promoters (Fig. 4D). Also, we found relatively higher levels of H3K4me1 and lower levels of H3K4me3 at enhancers occupied by CHD4 only and a relative loss of H3K4me1 and gain of H3K4me3 at sites occupied by GFI1 only and values for GFI1/CHD4 occupation in between (Fig. 4D). Analysis of RNA-seq data from GMPs of genes that are next to the promoters or enhancers defined here and were occupied by GFI1 showed higher expression levels than the genes next to sites bound by CHD4 or both CHD4 and GFI1 (Fig. 4E, F, respectively). These findings indicate that GFI1/CHD4 complexes occupy active or bivalent promoters and also active enhancers. These sites have a more closed chromatin configuration when CHD4 is present and have a more open chromatin conformation when GFI1 is present.

**De novo motif analysis highlights myeloid-specific gene network**. Next, we compared our ChIP-seq data for GFI1 and CHD4 binding and H3K4 methylation with a data set for the transcription factor CEBPα, also done in GMP cells, which has been shown to co-occupy promoter sites together with GFI1 in myeloid cells[42]. The data were again analyzed for promoter regions as defined above, they were ordered according to GFI1 binding and were separated into four groups: occupation by GFI1 alone, by GFI1/CEBPα, by GFI1/CEBPα/CHD4, and by GFI1/CHD4 (Fig. 5A). A de novo motif analysis[43] of consensus DNA binding sites at promoter regions revealed that the loci bound by GFI1 or by GFI1/CEBPα were very highly enriched for the GFI1/GFI1B DNA binding motif, as expected, but the enrichment was even higher for the consensus DNA binding motif for the transcription factor IRF1, which contains the 5′-AANNGAAA-3′ core sequence for all IRF factors (Fig. 5B). In addition, ETS2, STAT3, and SOX3 binding motifs were highly enriched at sites occupied by GFI1,

but in contrast, KLF, SPI1, RUNX, and ATF3 binding motifs were highly enriched at sites occupied by GFI1/CEBPα (Fig. 5B). At sites where GFI1 and CHD4 are both present, neither *Irf* nor *Gfi1* consensus sites were enriched. However, the binding motifs for SPI1 (PU.1) and the related factor SPI-C were found to be enriched with the highest *E* values (Fig. 5B). Sites occupied by GFI1/CHD4/CEBPα showed again enrichment for the PU.1 and SPI-C motifs, but in addition also the binding sequences for CEBPα and RUNX1 (Fig. 5B). This differential enrichment of binding motifs suggests that GFI1 or GFI1/CEBPα complexes bind to other genomic loci that those occupied by GFI1/CHD4 or GFI1/CEBPα/CHD4 complexes. Aggregation plots showed that H3K4me2 and H3K4me1 methylation levels at promoter regions occupied by GFI1 or GFI1/CEBPα complexes are differentially depleted or enriched at the TSS or 3′ of the TSS, respectively, according to the presence of CHD4 (Fig. 5C). The data suggest that GFI1 is more efficient in the depletion of H3K4me2 and -me1 marks in the presence of CHD4 (Fig. 5C). This also suggests that GFI1 and CEBPα are associated with promoters with a higher transcriptional activity than those where CHD4 is also present either together with GFI1 or GFI1/CEBPα complexes, and, in addition, suggesting again that the presence of CHD4 is associated with repressive histone marks.

**Chromatin remodeling by GFI1, CHD4, or both CHD4 and GFI1 during neutrophil differentiation**. The developmental steps from GMPs to mature neutrophils have been clarified using surface markers to define pre-neutrophils (preNeu), immature neutrophils (immNeu), and mature neutrophils (matNeu) stages[44]. To explore how sites that are occupied by GFI1, CHD4, or both in GMPs are altered during neutrophil differentiation with regard to chromatin openness or H3K4 dimethylation levels, we sorted GMP, preNeu and matNeu populations from bone marrow following a published strategy[44] (Suppl. Fig. 5A, Suppl. Fig. 12, Suppl. Table 5). Monitoring GFP expression in these same subsets isolated from *Gfi1*:GFP knockin mice[45] showed that the *Gfi1* gene is expressed and the promoter is active (Suppl. Fig. 5B, Suppl. Table 5). However, analysis by western blot of the sorted cells showed that GFI1 protein expression levels are higher in matNeus than in preNeu or immatNeu cells (Suppl. Fig. 5C, Suppl. Fig. 13, Suppl. Table 5). RNA-seq reads of groups of genes specific for chemotaxis or phagocytosis were obtained and were congruent with the published pattern (Suppl. Fig. 6A, B, Suppl. Table 5), similar to other genes regulated during myeloid differentiation indicating that the sorted populations represent indeed

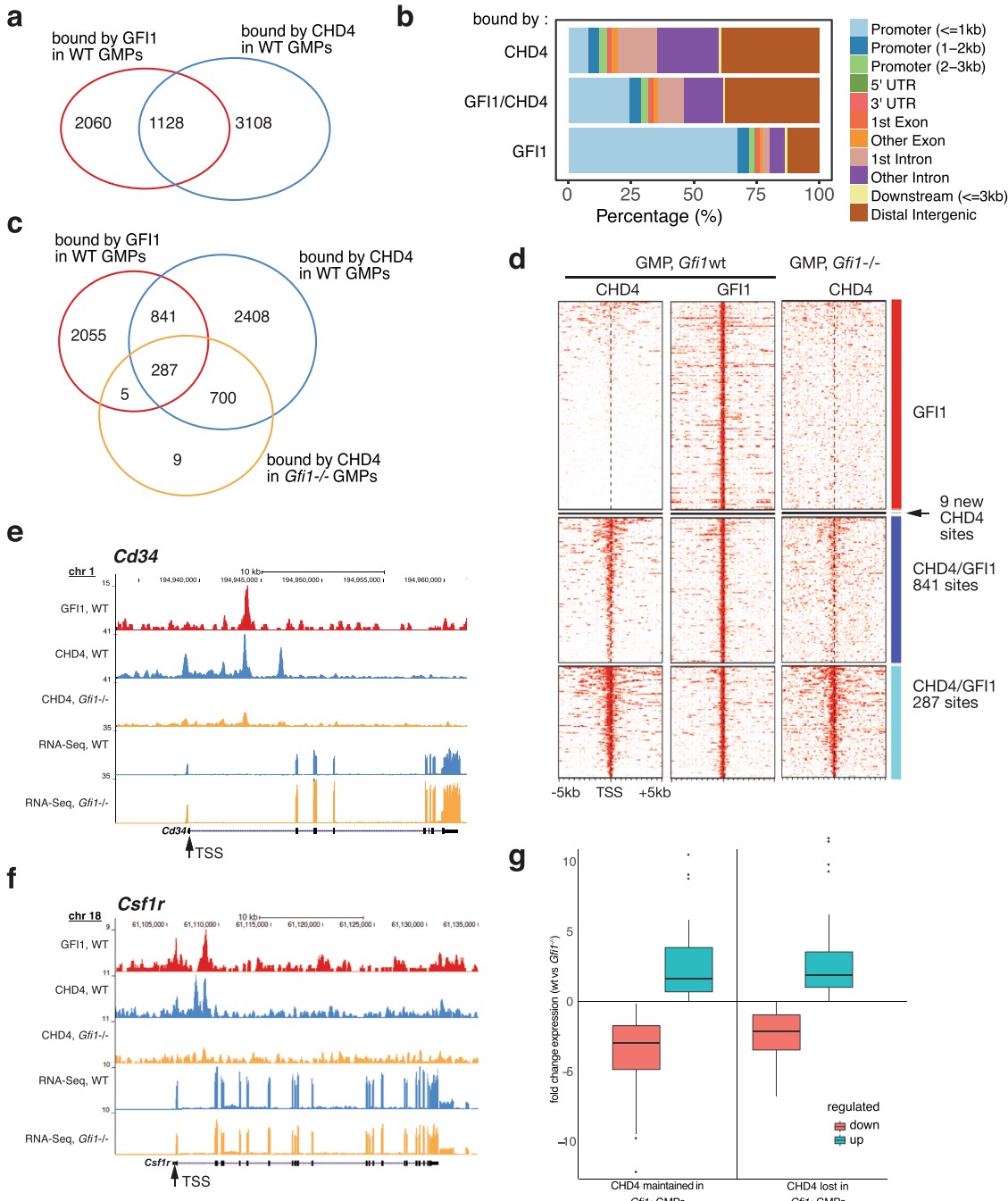

**Fig. 3 GFI1 and NuRD complex member CHD4 co-occupy sites at chromatin from granulocyte myeloid progenitors (GMPs). a** Venn diagram indicating the number of sites occupied by GFI1, CHD4, or both in GMPs according to ChIP-seq experiments done with GMPs. **b** Relative distribution of sites occupied by GFI1, CHD4, or both in percent of all sites in promoter regions, 3′ and 5′ untranslated regions (UTRs), introns, exons, or other distal regions. **c** Venn diagram indicating the number of sites occupied by GFI1 in WT GMPs, CHD4 in WT GMPs and CHD4 in GFI1 deficient GMPs. **d** Heatmap of ChIP-seq results for sites occupied by GFI1, or GFI1 and CHD4 in WT GMPs or in GMPs sorted from GFI1 deficient mice (*Gfi1* KO). Genes are sorted according to GFI1 occupation. Red: genes occupied by GFI1, but not by CHD4, dark blue: N = 841: sites occupied by both CHD4 and GFI1 in wt GMPs which lose CHD4 occupation when *Gfi1* is deleted (i.e., in *Gfi1* KO GMPs), pale blue: N = 287: sites occupied by both CHD4 and GFI1 in wt GMPs which maintain CHD4 occupation when *Gfi1* is deleted (i.e., in *Gfi1* KO GMPs). **e** Schematic depiction of the locus encoding murine CD34 and **f** the *Csf1r* locus encoding M-CSFR. Shown is the enrichment of reads after ChIP-seq with antibodies against GFI1 or CHD4 in GMPs from either WT or *Gfi1* KO mice and the enrichment of reads after an RNA-seq experiment from WT of *Gfi1* KO GMPs. The transcription start site is indicated (TSS, transcription start site). Annotations represent locations on the mouse genome version GRCm38 (mm10). **g** The promoters of the genes targeted by GFI1 and by CHD4 were separated into two groups according to the presence of CHD4 in *Gfi1* KO cells (w_CHD4 and wo_CHD4). The groups were then separated into two subgroups according to the direction of the fold change in expression after *Gfi1* deletion (up or down). For the comparison w_CHD4_up vs. wo_CHD4_up, the *p*-value is 0.80. For the comparison w_CHD4_down vs. wo_CHD4_down, the *p*-value is 0.0095. The lower and upper hinges correspond to the first and third quartiles (the 25th and 75th percentiles). The whiskers extend up to 1.5 times the inter-quartile range and data beyond this point were plotted individually.

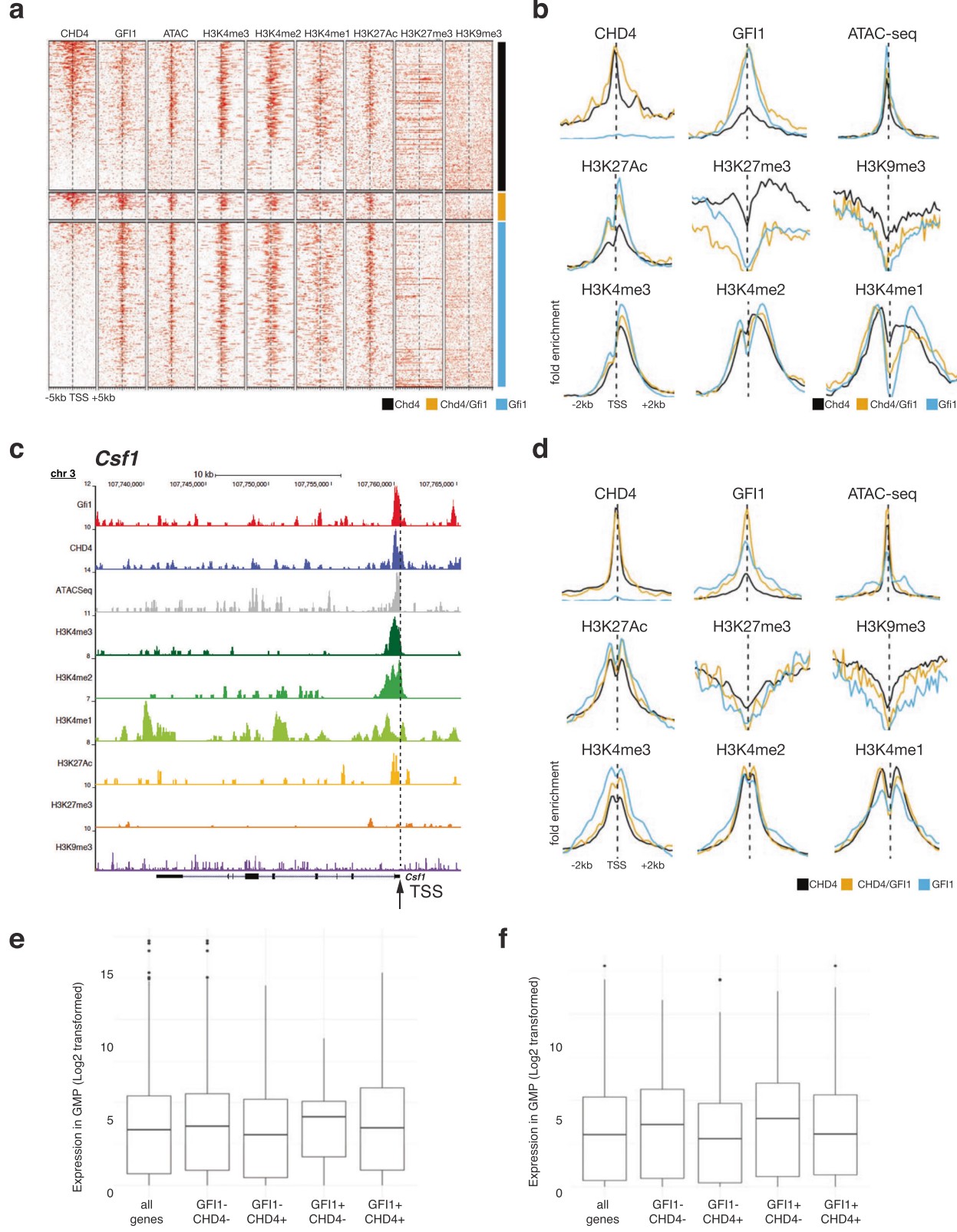

preNeu and matNeu cells as previously reported[44] (Suppl. Fig. 7, Suppl. Table 5).

Next, we performed both ATAC-Seq and ChIP-Seq experiments to determine whether and how chromatin openness and H3K4 di-methylation levels change at the loci that are occupied by GFI1 and CHD4 or both in GMPs during neutrophil differentiation. Data were filtered as described above for promoter

and enhancer regions, were ordered according to CHD4 occupation, and were separated into three groups: occupation in GMPs by CHD4 alone, by GFI1/CHD4, and by GFI1 alone (Fig. 6A, B, Suppl. Table 5). We compiled the data for the locus of a typical myeloid-specific enhancer localized in the 3′ region of the *PLBD1* gene[42] and several sites downstream (Fig. 6C, Suppl. Table 5). Sites 3′ of the *Plbd1* gene that are occupied by

**Fig. 4 GFI1 and GFI1/CHD4 occupy actively transcribed regions. a** Heatmap of ATAC-seq and ChIP-seq analyses were obtained with antibodies against GFI1, CHD4, and methylated and acetylated histone H3 from GMPs. Shown are reads for promoter regions with 5 kb 5′ and 3′ of TSS ordered according to CHD4 occupation and were separated into three groups: occupation by CHD4 alone, by GFI1 and CHD4, and by GFI1 alone. (TSS transcription start site). **b** Aggregation plots for promoters occupied by GFI1, CHD4, and methylated and acetylated histone H3 with the data shown in (**a**). Shown is the fold enrichment over a region of 2 kb 3′ and 5′ of the TSS. (TSS transcription start site). The differences in epigenetic marks for regions bound by CHD4 and or GFI1 in (**b**) and (**d**) (see below) are due to background signals since in some cases reads are counted which do not necessarily represent sites because cut off parameters were kept identical throughout the experiment. The signal in (**b**) and (**d**) represent the intensity of the reads that were mapped in the selected regions. To ensure all regions are compared between samples, the same parameters were used for the peak calling step in the same category. **c** Exemplary depiction of the ATAC-Seq and ChIP-Seq data at the GFI1 target gene *Csf1*. Indicated are the tracks corresponding to the individual experiments, the gene, and the TSS. Annotations represent locations on the mouse genome version GRCm38 (mm10). **d** Aggregation plots for enhancers occupied by GFI1, CHD4, and methylated and acetylated histone H3. Shown is the fold enrichment over a region of 2 kb 3′ and 5′ of the TSS (transcription start site). **e** Analysis of RNA-seq data from GMPs of genes that are next to the promoters or **f** enhancers and were occupied by GFI1, CHD4, or both CHD4 and GFI1. The lower and upper hinges correspond to the first and third quartiles (the 25th and 75th percentiles). The whiskers extend up to 1.5 times the interquartile range and data beyond this point were plotted individually.

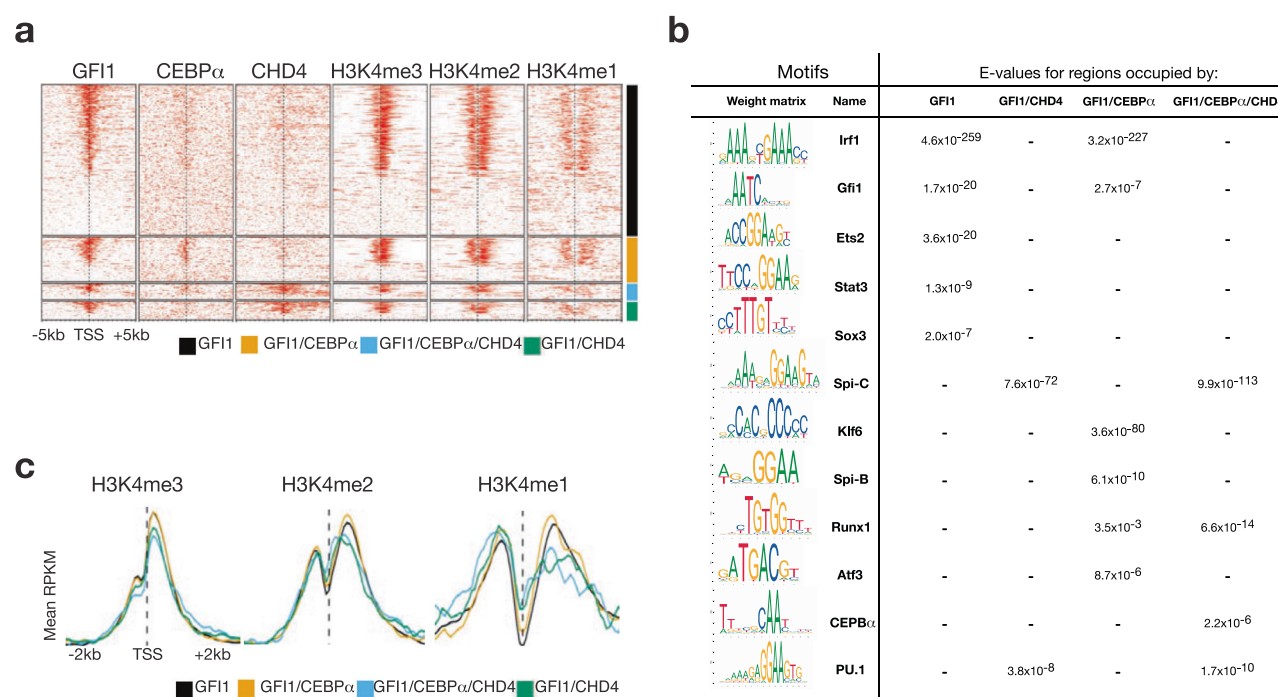

**Fig. 5 De novo motif analysis and histone methylation aggregation plots for sites occupied by GFI1, CHD4, and CEBPα in GMPs. a** Comparison of ChIP-seq data for the occupation of GFI1, CHD4, and H3K4 methylation at promoter regions in GMPs with an analogous ChIP-seq data set for the transcription factor CEBPα. Data are ordered according to GFI1 occupation and are separated into four groups: occupation by GFI1 alone, by GFI1/CEBPα, by GFI1/CEBPα/CHD4, and by GFI1/CHD4. **b** De novo motif analysis of consensus DNA binding sites at promoter regions by GFI1 alone, by GFI1/CEBPα, by GFI1/CEBPα/CHD4, and by GFI1/CHD4. **c)** Aggregation plots of histone H3K4 methylation at promoters as defined in (**a**) for regions 2 kb 5′ or 3, of the TSS (transcription start site).

GFI1/CHD4 in GMPs showed a decrease in ATAC-seq reads and low-level H3K4me2 marks compared to the site within the *Plbd1* gene where CHD4 is present without GFI1. At this site, ATAC-seq levels increased and H3K4me2 levels accumulated in preNeu and matNeu cells (Fig. 6C, Suppl. Table 5).

To better quantify and facilitate the integration of signals from GMPs, preNeu, und matNeu cells of regions that are occupied by GFI1, CHD4, or both GFI1 and CHD4 in GMPs, we performed a Metagene analysis[46]. This allowed us to directly compare the enrichment profiles of H3K4me2 and ATAC-seq signals between cell types and experiments and to visualize the result. We used the same definition of promoter and enhancer regions as in the previous analyses to extract the enrichment signal, which was then normalized using the NCIS algorithm[47]. This permitted us to compare the mean coverage values between cell types (GMP, preNeu, matNeu), since the normalization integrates both background noise and the size of the library (RPM). Bins cover 1000 base pairs 5′ and 3′ of the TSS for the promoter region and of the defined center region (Ctr) of the enhancer (at 50 bins).

Using the ChIP-seq data from GMPs, we compiled values from 2190 promoter sites occupied by GFI1, 1779 sites occupied by CHD4, and 341 sites occupied by both CHD4 and GFI1. We observed that mean coverage of ATAC-seq RPM values for these promoters was around 15-fold lower in preNeu and matNeu cells than in GMPs (Fig. 6D, Suppl. Table 5). Also, within preNeu and matNeu cells, they were significantly lower at sites that were occupied by CHD4 than those occupied by GFI1 in GMPs ($p < 10^{-30}$, red lines Fig. 6D, Suppl. Table 1), indicating that chromatin regions close during neutrophil differentiation and that sites occupied in GMPs by CHD4 are more contracted than sites occupied in GMPs by GFI1 or GFI1/CHD4. Similarly, at the same sites, the mean coverage of H3K4me2 levels were lower overall in preNeu and matNeu cells compared to GMPs (Fig. 6D, Suppl. Table 5). They were also lower at the promoter sites

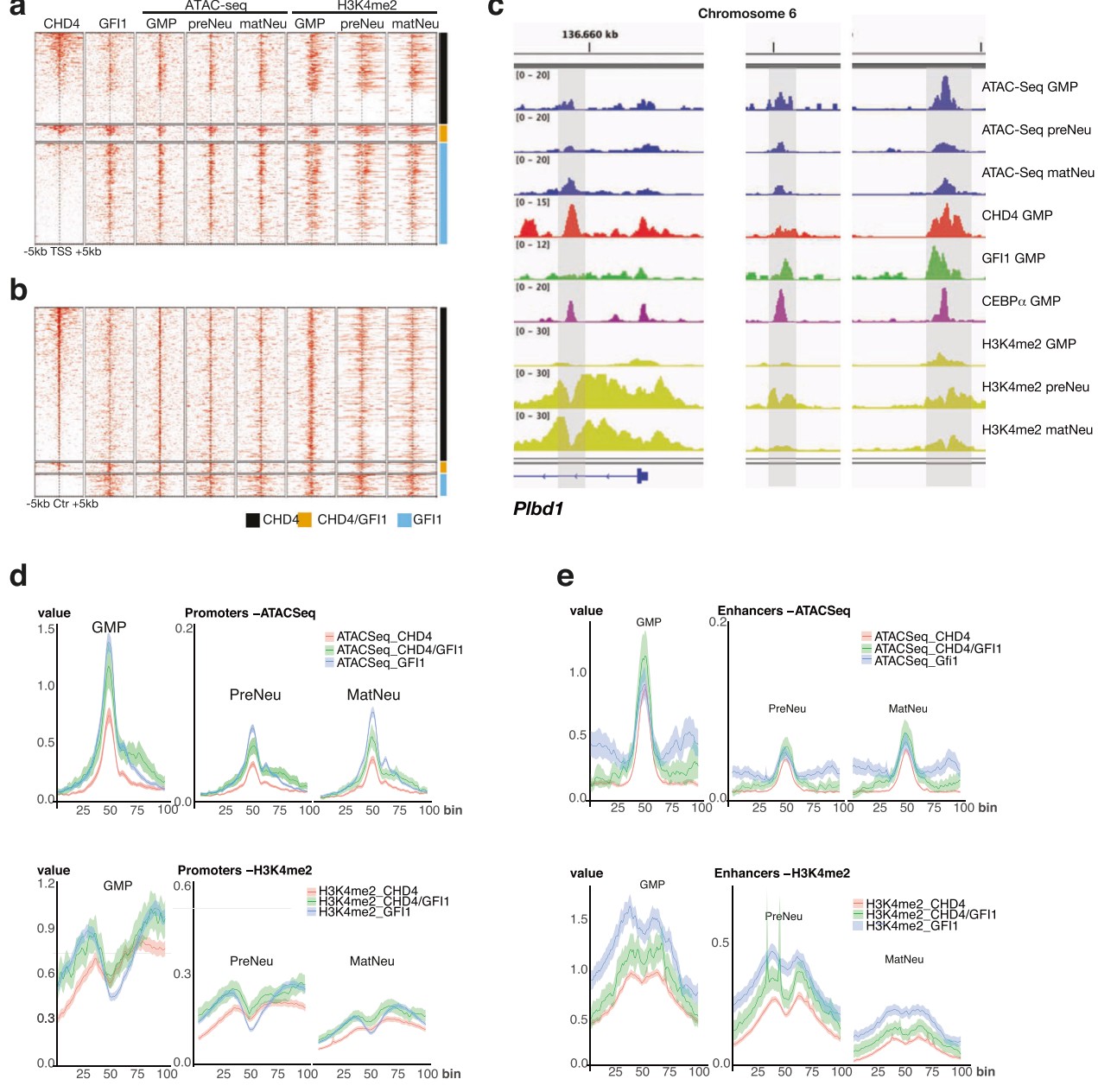

**Fig. 6 H3K4 methylation and chromatin status at sites occupied by GFI1, CHD4, or both CHD4 and GFI1 during neutrophil differentiation. a** Heatmap of data from ChIP-Seq and ATAC-Seq experiments and to determine the occupation of GFI1, CHD4, and H3K4 di-methylation patterns during neutrophil differentiation from GMPs via preNeu (preneutrophils) to matNeu (mature neutrophils). Data were filtered for promoter regions, ordered according to CHD4 occupation, and were separated into three groups: occupation by CHD4 alone, by GFI1/CHD4 and by GFI1 alone. **b** Heatmap of data from ChIP-Seq and ATAC-Seq experiments as in (**a**), but for enhancer regions. **c** ATAC-Seq and ChIP-seq read on loci covering the myeloid-specific enhancers at the 3′ end of the *Plbd1* gene and at downstream regions for sites occupied by GFI1, CHD4, or both, CEBPa and carrying the indicated histone marks. Annotations represent locations on the mouse genome version GRCm38 (mm10). **d** Metagene analysis for ATAC-seq reads and H3K4me2 levels (**e**) in RPM values for GMPs, preNeu, and matNeu cells at promoters occupied by CHD4 alone, by GFI1/CHD4, and by GFI1 alone. Shown are regions 2 kb 5′ and 3′ of the TSS. **e** Metagene analysis for ATAC-seq reads and for H3K4me2 levels in mean RKPB for GMPs, preNeu and matNeu cells at enhancers occupied by CHD4 alone, by GFI1/CHD4, and by GFI1 alone. Shown are regions 2 kb 5′ and 3′ of the site of the enhancer (Ctr).

occupied by GFI1 compared to sites occupied by CHD4 or GFI1/ CHD4 complexes at the TSS (blue line Fig. 6D, Suppl. Table 2, Suppl. Table 5), suggesting that GFI1 may be active in removing methyl groups from H3K4 in GMPs and preNeu cells and that this is modified when CHD4 is present together with GFI1.

For enhancers, we included values from 382 sites occupied by GFI1, 2375 sites occupied by CHD4, and 173 sites occupied by both CHD4 and GFI1 in GMPs. The RPM values of ATAC-seq reads indicated a highly significant chromatin contraction for

these sites in preNeu and matNeu cells versus GMPs (Fig. 6E, Suppl. Data 1, Suppl. Table 5), but no differences were seen in each cell type between enhancer centers that were occupied by GFI1, CHD4, or GFI1/CHD4 in GMPs, only at regions 5′ and 3′ of the center when occupied by GFI1 (Fig. 6E, Suppl. Data 2, Suppl. Table 5). H3K4me2 values at these enhancers were again lower in preNeu and matNeu cells than in GMPs (Fig. 6E, Suppl. Data 2, Suppl. Table 5). However, sites at the enhancer centers occupied in GMPs by CHD4 had significantly lower H3K4me2

levels than sites occupied by GFI1 (Fig. 6E, red lines and blue lines, respectively, Suppl. Table 5), whereas sites occupied by GFI1/CHD4 had intermediate levels (Fig. 6E, green line, Suppl. Data 2, Suppl. Table 5). This indicates that in GMPs GFI1 and CHD4 act differently on H3K4 methylation at enhancer sites than at promoters and that this differential pattern is retained in preNeu and matNeus cells.

**Genes occupied by CHD4, GFI1, or both belong to the different categories**. Next, we reordered the Chip-Seq and ATAC-seq data once again according to CHD4 occupation at promoter and enhancer sites but now separated them into 6 groups: occupation by CHD4 alone, by GFI1/CHD4, or by GFI1 alone and according whether these groups of genes were up or down-regulated during the differentiation from GMPs to preNeu and matNeus (Fig. 7A, B, Suppl. Table 5). A GO classification showed that genes occupied by CHD4 alone are found in pathways typical for the immune and inflammatory response regardless of whether they were upregulated or downregulated during differentiation from GMPs to matNeu cells (Fig. 7B, Suppl. Table 5). Up and downregulated genes co-occupied by both CHD4 and GFI1 encode regulators of chromatin assembly and nucleosome organization, while genes occupied by GFI1 alone are involved in metabolic processes (Fig. 7C, Suppl. Table 2, Suppl. Table 5), suggesting that GFI1, CHD4, and the GFI1/CHD4 complex regulate separate defined groups of genes during neutrophil differentiation.

## Discussion

In the present study, we describe results of BioID experiments indicating that the SNAG domain and zinc-finger transcriptional repressor GFI1 associates with several chromatin remodeling complexes such as the NuRD complex, which also contains LSD1 and members of the LSD1/CoREST complex such as RCOR1, -2 and -3, but also the CtBP and SWI/SNF chromatin remodeling and repressor complexes. We chose to elucidate the biological meaning of the interaction with the NuRD complex since we find that GFI1 can associate with almost all its components. We have focused on the chromodomain helicase CHD4, one of the NuRD complex components, and have used primary murine cells representing stages of neutrophil differentiation as a model system given the particularly important role of GFI1 in this process. We demonstrate that GFI1 can recruit CHD4 to specific sets of target genes that regulate processes such as nucleosome organization and chromatin assembly. While GFI1 occupies target gene promoters containing its own cognate DNA consensus sequence and those for IRF1, GFI1/CHD4 complexes target genes at consensus sites for ETS-related factors such as SPI1 (PU.1) and SPIC. Analysis of histone modifications and chromatin structure indicates that GFI1 and GFI1/CHD4 complexes occupy active or bivalent promoters and active enhancers, both up and down-regulated during neutrophil differentiation.

Previous studies showed that GFI1 binds to LSD1 and HDACs[1,9,48], but these histone-modifying enzymes are also members of the NuRD complex. It is thus possible that GFI1 enters into association with the NuRD complex either via LSD1 or HDACs or through CHD4 or MTA2, the proteins identified here to bind to GFI1. Immunoprecipitation and mass spectrometry data, BioID experiments with full-length GFI1 and a GFI1 mutant that lacks the SNAG domain, biochemical analyses with mutated and truncated GFI1 proteins support a model in which GFI1 associates with the CHD4 and MTA2 components of the NuRD complex through several regions including the SNAG and zinc finger domains. Given that the SNAG domain specifically and directly interacts with LSD1[1,8], it is possible that GFI1

associates with LSD1 and at the same time with components of the NuRD complex such as CHD4 and MTA2. SEC fractionation data support this view, since the GFI1 complexes that elute around 400–500 kDa contain LSD1 and RCOR1, but also CHD4 and MTA2. Data from the SEC however also indicate that GFI1 is a member of other, high molecular weight complexes eluting at around 2 MDa, in which NuRD components or members of the LSD1/CoREST complex are much less abundant. This would be in agreement with the notion that GFI1 acts as a member of several different chromatin remodeling complexes, which is also supported by our BioID results. The precise nature of the 2 MDa GFI1 complex, however, remains to be determined.

Since we can demonstrate that GFI1 and CHD4 co-occupy promoter and enhancer sites in GMPs, it is conceivable that GFI1 recruits the NuRD complex to these regions in the chromatin. For a considerable fraction of these sites, the recruitment is likely to occur directly through GFI1, since the deletion of GFI1 abrogates CHD4 occupation. The deletion of GFI1 did not lead to a large redistribution of CHD4 to new sites as it was observed in similar experiments with the transcription factor IKAROS in B lymphocytes[33], which underlines an important difference to our study with myeloid cells. According to the existing model, sites occupied by GFI1 should be depleted of H3K4me2, -me1, and H3K9acetyl marks and be transcriptionally silent[1,49]. However, our ChIP-seq data suggest that GFI1 and CHD4 and GFI1/CHD4 complexes are located at transcriptionally active promoters in GMPs. The enrichment of markers for active transcription such as acetylation at H3K27 and di- and tri-methylation at H3K4 and the depletion of markers indicating transcriptional repression such as tri-methylation of H3K27 and H3K9 at sites where GFI1 binds support this notion. However, while H3K4me1 that indicates active transcription is strongly depleted at the TSS of GFI1 occupied genes, it is possible that these are rather bivalent promoters, which are characterized by the presence of both active and repressive histone marks.

Similarly, enhancers occupied by GFI1, CHD4, and GFI1/CHD4 complexes show H3K4me1 and H3K27 acetylation and absence or depletion of H3K27me3 or H3K9me3 marks, which is consistent with active enhancers as opposed to inactive or poised enhancers, which would be characterized by the presence of both H3K4me1 and H3K27me3. Of interest here is that sites occupied by CHD4 show relatively lower levels of active histone marks and relatively higher levels of histone marks associated with inactive promoters and enhancers compared to sites occupied by GFI1 alone. Sites occupied by both GFI1 and CHD4 consistently show levels in between suggesting that GFI1 can modulate the effect of the NuRD complex on chromatin remodeling. RNA-seq data show that genes associated with promoters or enhancers occupied only by GFI1 are expressed at a higher level than genes occupied only by CHD4, while genes associated with promoters or enhancers occupied by both GFI1 and CHD4 have expression levels that fall in between these extremes. This supports the view that GFI1 is associated with active promoters and enhancers and can modulate the effect of the NuRD complex. This suggests also that the presence of the NuRD complex at sites occupied by GFI1 reduces transcription relative to sites occupied by GFI1 alone.

The active chromatin found here at sites occupied by GFI1 and GFI1/CHD4 complexes in progenitor cells such as GMPs seems at first sight to contradict the established function of both GFI1 and the NuRD complex as transcriptional repressors. In particular, H3K4me2 levels were expected to be depleted at GFI1-occupied sites given its association with LSD1, but this was not always observed when comparing values from aggregation plots. However, it is conceivable that GFI1 or GFI1/NuRD complexes are in a poised state in GMPs and become active only upon differentiation signals that enable them to repress the transcription of genes specific for neutrophil

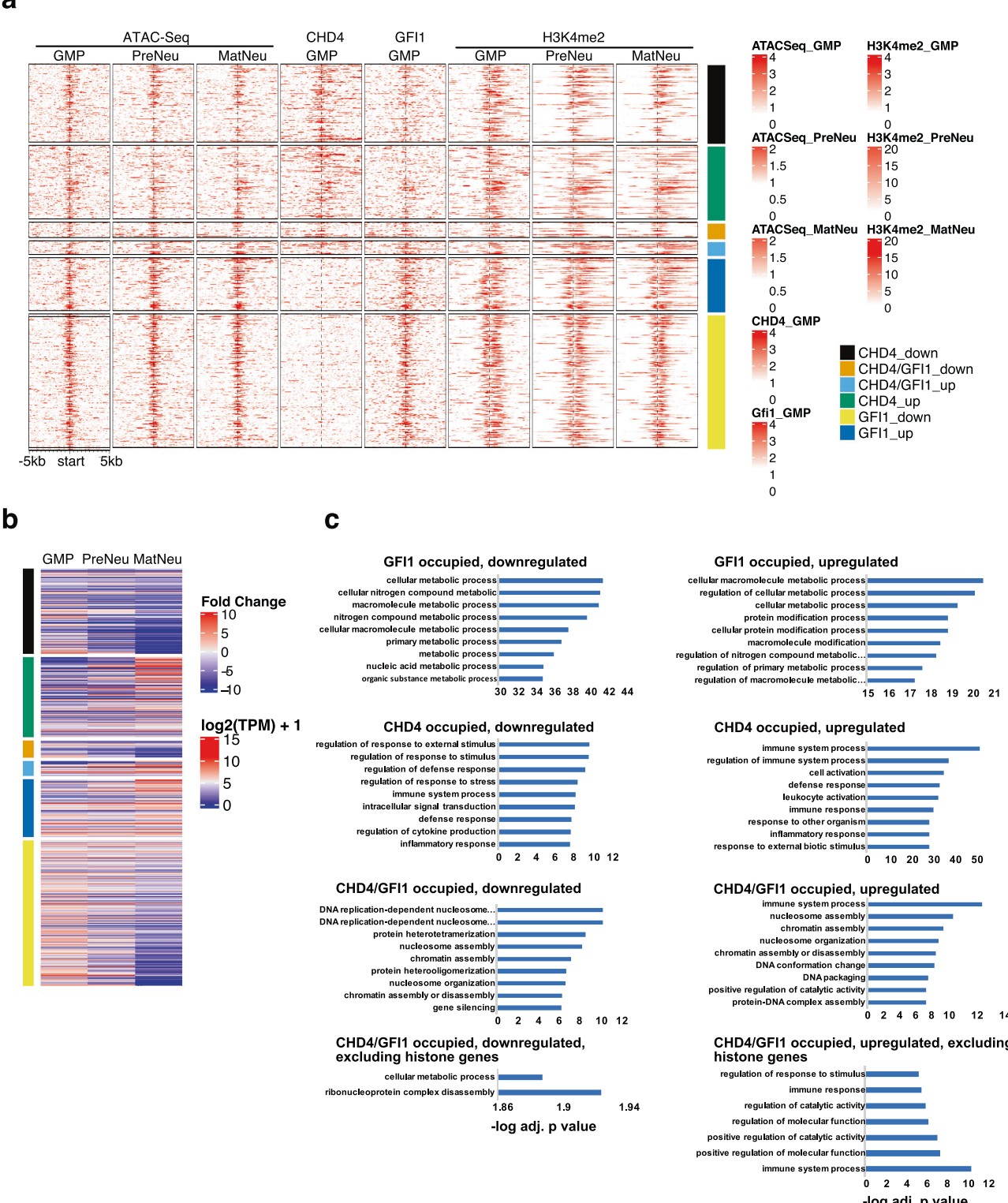

differentiation. Aggregation plots from ATAC-seq analyses suggest more accessibility at promoter sites at a narrow region around the TSS compared to the enhancer sites. However, the pattern of the ATAC-seq plot is different at enhancers where it is broader compared to promoters, where it is narrower. This does not allow to conclude that promoters are more accessible than enhancers but

rather indicates that different mechanisms are probably in place controlling accessibility at enhancer versus promoter sites occupied by GFI1.

Our Metagene analyses, which allowed a quantification and direct comparison between cell types and experiments, provided some clarification and indicated that chromatin openness and H3K4me2

**Fig. 7 Chromatin status and H3K4 dimethylation pattern during neutrophil differentiation. a** Heatmap of Chip-Seq and ATAC-seq data obtained in GMPs, preNeu, and matNeu cells ordered according to CHD4 occupation at promoter sites in GMPs. The data were separated into six groups: occupation by CHD4 alone, by GFI1/CHD4 complexes, or by GFI1 alone and according to whether these groups of genes were up or down regulated during the differentiation from GMPs to preNeu and matNeu cells. Scale bars indicate read coverage. **b** Heatmap of RNA-Seq data from GMPs, preNeu, and matNeu cells, ordered according to genes with promoters occupied by CHD4 alone, by GFI1/CHD4 complexes or by GFI1 alone and according whether these groups of genes were upregulated or downregulated during the differentiation from GMPs to preNeu and matNeu cells. **c** GO classification of genes occupied by CHD4 alone, by GFI1/CHD4 complexes, or by GFI1 alone and according to whether these groups of genes were up or downregulated during the differentiation from GMPs to preNeu and matNeu cells. Examples of genes for common pathways are given in Suppl. Table 2.

levels at sites occupied by GFI1, CHD4, and GFI1/CHD4 complexes are strongly reduced when GMPs differentiate into preNeu and matNeu cells. In addition, chromatin was more compacted at enhancers bound by CHD4 than at those occupied by GFI1 or GFI1/CHD4. Interestingly, however, H3K4me2 patterns were lowest at the TSS of promoters occupied by GFI1 alone relative to sites occupied by CHD4 or GFI1/CHD4 complexes, which would be in agreement with the presumed function of GFI1 to enable the removal of methyl groups from H3K4 via LSD1. At enhancers, however, H3K4me2 levels are highest at sites occupied by GFI1 compared to sites bound by CHD4 or both GFI1 and CHD4, indicating that the role of GFI1 as a facilitator of chromatin remodeling may be different at promoters and enhancers.

Previous studies have shown that a regulatory network exists between the transcription factors PU.1, IRF8, C/EBPα, and GFI1 in myeloid differentiation and that GFI1 can bind to regions in chromatin together with PU.1 or C/EBPα or IRF8[49–51]. Evidence from our motif analysis suggests that CHD4 affects the role of GFI1 in this regulatory network. While GFI1 and GFI1/C/EBPα complexes bind to sites similarly enriched for IRF and GFI1 consensus binding sites, complexes that contain CHD4 are found at sites enriched for PU.1 binding sites but lack a GFI1 consensus site. This suggests that the presence of CHD4 redirects GFI1 to a different set of promoters, or at least to a different region of a promoter. This also points to the possibility that when in a complex with CHD4, GFI1 may not be required to directly contact DNA or does so through a component of the NuRD complex such as MTA2, which has a DNA binding domain[30]. The analysis of genes up and downregulated during neutrophil differentiation from GMPs via preNeu to matNeu cells supported this view since it showed that genes occupied by CHD4, GFI1, or GFI1/CHD4 complexes belong to different groups. We propose that in myeloid progenitors GFI1 tethers the NuRD complex through the binding to CHD4 and other components to a specific set of target genes with active or bivalent promoters and active enhancers but remains at a poised state. During neutrophil differentiation, CHD4, GFI1, and GFI1/CHD4 complexes enable the transcriptional regulation of different sets of target genes affecting the immune response, cellular metabolic processes, or nucleosome organization through chromatin remodeling. It has been reported that the deletion of *Chd4* leads in mice to the reduction of neutrophil granulocytes and B-cell precursors[52], which are the same cells that require GFI1, and to an increase in erythroid precursors, which are known to require GFI1B. Although the biological role of the interaction of GFI1 and GFI1B with members of the NuRD complex remains to be investigated in more detail, and the absence of an in vivo model such as the use of *Chd4* gene-deficient mice represent a limitation of the present study, our findings suggest that the association of GFI1 and GFI1B with the NuRD complex is important for all early hematopoietic lineages, notably for the development of the myeloid lineage.

## Methods
**Mouse strains.** The mice used in this study have been bred on to C57BL/6 genetic background and were maintained in a specific-pathogen-free plus environment at the Institut de recherches cliniques de Montréal (IRCM). The Institutional Animal Ethics Review Board of the IRCM approved all animal protocols and experimental procedures were performed in compliance with IRCM and CCAC (Canadian

Council of Animal Care) guidelines. The work was performed under the IRCM identification number 2020-09 TM.

**BioID and MS data analysis.** BioID experiments were carried out similarly to those in previous reports, with modifications[37,53,54]. Briefly, Flp-In T-REx HEK293T cell lines from 15-cm plates expressing GFI1WT-BirA*-Flag, GFI1ΔS-NAG-BirA*-Flag in a tetracycline-inducible manner were treated with 50 μM biotin and 1 μg/ml tetracycline for 24 h. The following day, the medium was removed, and cells were washed three times with cold phosphate-buffered saline (PBS). Cell pellets were lysed in 1.5 ml radioimmunoprecipitation assay (RIPA) buffer [50 mM Tris-HCl (pH 7.4), 150 mM NaCl, 1% NP-40, 1 mM EDTA, 0.1% SDS, 0.5% sodium deoxycholate, Sigma protease inhibitor 1:500 (P8340-5ml, Sigma), 1 mM DTT, 1 mM PMSF, supplemented with 250 U benzonase (71205-3, EMD Millipore). The samples were sonicated at 30% amplitude for 30 sec (three 10 sec bursts with 2 s pauses) at 4 °C. Cleared cell lysates were incubated with 70 μl of pre-washed streptavidin–sepharose beads (17-5113-04, GE Healthcare) on a rotator for 3 h at 4 °C. Beads were then washed with 1 ml RIPA buffer, transferred to a fresh Eppendorf, and washed two times with 1 ml RIPA buffer. Beads were then washed three times in 1 ml of 50 mM ammonium bicarbonate (ABC) (AB0032, Biobasic). Samples were then subjected to reduction/alkylation with TCEP and iodoacetamide, and then trypsin digestion by the IRCM proteomics core facility. Trypsin-digested samples were resuspended in 2% acetonitrile prior to mass spectrometry analysis. Samples were analyzed by Orbitrap Fusion Mass Spectrometer (Thermo Fisher) at the IRCM proteomics core facility. Peptide searches and protein identification analyses for GFI1 WT-BirA*-Flag or GFI1 ΔSNAG-BirA*-Flag samples were performed similarly to those in previous reports[36,37,53–56]. BirA*-Flag controls and GFI1BWT-BirA*-Flag samples were generated in our previous work[37]. BioID-MS data were processed using the ProHits software[57,58]. The Proteowizard4 tool was used to convert RAW files to.mzXML files. MS/MS spectra were searched by using Human RefSeq version 57. Peptide search was carried out through Mascot and X!Tandem[59,60]. The Trans-Proteomic Pipeline (TPP) suite and iProphet integrated into ProHits were used for peptide validation and quantitation[55,60].

**Dot plot analysis.** Significance Analysis of INTeractome (SAINT) implemented in ProHits was used to score protein interactions based on the average number of spectral counts and for data filtering. In brief, SAINT files of the wild-type GFI1 (GFI1WT-BirA*-Flag) and GFI1-ΔSNAG (GFI1ΔSNAG-BirA*-Flag) and the wild-type GFI1B (GFI1BWT-BirA*-Flag) were filtered using BirA*-Flag controls. SAINT output files of GFI1/GFI1B (Fig. 1C), and GFI1/GFI1ΔSNAG (Fig. 2D) generated in ProHits were processed by using the ProHits-viz tools to carry out dot plot analyses[37,57,58,61]. Dot plots displaying the protein interaction data included the average number of spectral counts for each prey (AvgSpec), the relative prey abundance towards baits (relative abundance), and the Bayesian false discovery rate (BFDR) as confidence score for the indicated interacting proteins.

**Protein network analysis.** Protein network analyses were performed by using Cytoscape[62]. The SAINT output file of GFI1-BirA*-Flag BioID/MS data was imported to Cytoscape. The existing protein–protein interaction network between preys identified in GFI1-BirA*-Flag BioID/MS screen was imported by using the Biogenet network analysis tool and merged with the GFI1-BirA*-Flag BioID/MS data[63]. The merged protein network was then subjected to MCODE clustering to visualize protein complexes that are connected with the GFI1 bait[64].

**Preparation of nuclear extracts (NE) for SEC.** Kasumi cells (2.5 billion) were harvested from cell culture media by centrifugation (at RT) for 10 min at 2000 rpm. Divided into two 50 mL falcons. Pelleted cells were suspended in five volumes of 4 °C PBS and collected by centrifugation; subsequent steps were performed at 4 nuclear extracts. The cells were suspended in five packed cell pellet volumes of buffer A (10 mM HEPES (pH 7.9), 1.5 mM MgCl2, 10 mM KCl, 0.5 mM DTT, 0.5 mM PMSF) and allowed to stand for 10 min. The cells were collected by centrifugation as before and suspended in two packed cell volumes (volume prior to the initial wash with buffer A) of buffer A and lysed by 15 strokes of a Dounce Homogenizer (B type pestle). The homogenate was checked microscopically for cell lysis and centrifuged for 10 min at 2000 rpm to pellet nuclei. The pellet obtained from the low-speed centrifugation of the homogenate was subjected to second

centrifugation for 15 min at 25,000$g$ (Beckman Coulter JA-25.15 Fixed Angle Rotor), to remove residual cytoplasmic material and this pellet was designated as crude nuclei. To extract the nuclear protein, 1.3 ml of IP lysis buffer (50 mM Sodium phosphate, 300 mM NaCl, 1 mM beta-ME (add fresh before use), 10% glycerol, 0.5% NP40, 0.5% Triton-x100, pH7.5) containing protease inhibitors (PIC: 40 µl/ml and PMSF: 0.5 mM) were added to the pellet, kept on ice for 15 min and vortexed every 2 min. Protein-DNA complex was sonicated 3 times for 10 s at 50% output using the Brason digital sonifier, once for 15 s at 50% output then put on ice for 10 min and centrifuged at 13,000 rpm for 10 min at 4 °C. Totally, 3.25 mL of supernatant (the equivalent of $2.5 \times 10^9$ cells) and aliquoted into 250 µl (12 tubes) and 50 µl (5 tubes) at a concentration of 2.6 µg/µl.

**Size exclusion chromatography**. Prior to SEC, 200 µl of NE were cleared by centrifugation at 16,000×$g$ for 10 min at 4 °C and was size-fractionated on a Superose 6 10/300 SEC column connected to an AKTA-Purifier 10 (Cytiva). Prior to sample fractionation, the column was first calibrated using the high molecular weight gel filtration calibration kit (Cytiva). Before injection onto the column, the NE aliquot was cleared by centrifugation at 16,000×$g$ for 10 min at 4 °C. Isocratic elution was carried out in 50 mM sodium phosphate pH 7, 150 mM NaCl, 10% glycerol, 0.5% NP40 and 0.5% Triton-X100, and 500 µl fractions were collected. For western blotting analysis, each fraction was precipitated by a TCA/DOC method adapted from the literature[65]. To every collected fraction, 4.25 µl of 2% DOC was added and vortexed before incubating the samples on ice for 30 min. One-tenth volume of 100% TCA was then added to the fraction, vortexing immediately upon addition of TCA. Samples were left on ice for another 30 min and proteins were recovered by centrifugation at 16,000×$g$ for 15 min at 4 °C. Protein pellets were washed twice with 800 µl of −20 °C acetone, air dried, and resolubilized in 20 µl of 2× LDS sample loading buffer (BioRad) for two hours at 37 °C, with the last hour under agitation (1000 rpm) using a ThermoMixer (Eppendorf). DTT was added to the protein fractions that were further denatured for 10 min at 70 °C before loading and separating 75% of the material along with 5 µl of NE as input on 15-wells 4–15% Mini Protean TGX gels (BioRad). Proteins from the gel were transferred on a PVDF membrane overnight at 4 °C and 30 v in Towbin buffer containing 10% of methanol. The membranes were blocked in 5% non-fat milk proteins, and proteins of interest were probed with corresponding antibodies.

**GO term and CORUM analyses**. Gene Ontology (GO) and the comprehensive resource of mammalian protein complexes (CORUM) analyses were carried out similarly to those in previous reports using the g:Profiler tool[37,66,67]. Briefly, molecular function, biological process or CORUM protein complex analysis of prey interactions recovered in GFI1WT-BirA*-Flag or GFI1ΔSNAG-BirA*-Flag BioID-MS screens are illustrated in heatmap analyses. Contaminant proteins such as non-specific interactions or false positives were filtered by using the Contaminant Repository for Affinity Purification (CRAPome) repository prior to the GO term analysis[68]. Reviewed UniProtKB entry for each prey protein analyzed in Significance analysis of INTeractome (SAINT) file from ProHits were searched in g:Profiler for GO term or CORUM analysis[61,67]. GO term enrichment scores were calculated based on the −log10 of corrected $P$ values.

**Metagene analysis**. The metagene profile of the enrichment of the ATACSeq and H3K4me2 ChIP-Seq experiments in GMP, preNeu, and matNeu cell types were produced with previously described bioinformatic procedures[47]. The signal was normalized using the NCIS algorithm[46]. The metagene profile of the enrichment of the ATACSeq and H3K4me2 ChIP-Seq experiments in GMP, preNeu, and matNeu cell types were produced using the metagene2 package version 1.0.0[46]. Briefly, the signal was extracted from the alignment files in the promoter or enhancer regions targeted by each factor, the alignments were converted into coverage which were normalized using the NCIS algorithm[47] and plotted using built-in functionalities from the metagene2 package.

**Flow cytometry analysis, sorting of GMPs**. Hematopoietic cells were analyzed with LSR, or LSR Fortessa flow cytometer (BD Biosciences, Mountain View, CA) and analyzed using BD FACS Diva software (BD Biosciences) or FlowJo (for histogram analysis; Tree Star). For cell sorting, lineage negative BM cells were first depleted using a mouse lineage cell depletion kit (Miltenyi Biotec) then applied to a five-laser FACSAria II sorter (BD Biosciences) (Suppl. Figs. 11 and 12).

**Cell culture**. THP-1 (ATCC TIB-202), KASUMI (ATCC CRL-2724) cells were maintained in RPMI media (Multicell) supplemented with 10% bovine growth serum (RMBIO Fetalgro) and 100 IU Penicillin and 100 µg/ml Streptomycin (Multicell). HEK293T (ATCC CRL-1573) cells were maintained DMEM media (Multicell) with the above-mentioned supplements. We verified that none of the cell lines used in this study were found in the Register of Misidentified Cell Lines maintained by the International Cell Line Authentication Committee. All cell lines used in this study were tested and shown to be negative for mycoplasma contamination using PCR amplification using a mix of primers (https://bitesizebio.com/23682/

homemade-pcr-test-for-mycoplasma-contamination/, 2015), also shown in the list of oligonucleotides (Suppl. Table 3).

**Western blots**. Uncropped and unprocessed scans of the western blots are provided in the Supplementary material (Suppl. Figs. 8–10).

**Gene expression profiling by RNA-seq analysis**. Bone marrow from 2 tibiae, 2 femora, and 2 humeri was harvested in PBS/2.5% FBS and pooled before lineage negative depletion using autoMACS Pro separator (Miltenyi Biotec). Cells were incubated with a lineage antibody cocktail (B220, CD3, CD4, CD8, GR1, CD11b, NK1.1, Il7R, CD19, Suppl. Table 4) and were sorted on FACSAria II sorter (BD Biosciences) to recover GMPs. RNA was extracted using MagMax-96 Total RNA Isolation kit (Ambion) and quality-checked with RNA 6000 Pico kit (Agilent). RNA-seq libraries were prepared from the RNA extracts using the Illumina TruSeq Stranded mRNA Kit according to the manufacturer's instructions and sequenced using the TruSeq PE Clusterkit v3-cBot-HS on an Illumina HiSEq 2000 system. Sequencing reads were aligned to the mm10 genome using Tophat v2.0.10[69]. Reads were processed with Samtools[70] and then mapped to Ensembl transcripts using HTSeq[71]. Differential expression was tested using the DESeq R package[72] (R Core Team 2015, http://www.r-project.org/). A genome coverage file was generated and scaled to RPM using Bedtools[73]. RNA-seq data produced for this study are available under accession number GSE173533. RNA-Seq samples are in triplicate.

**Functional analysis**. The enrichment of selected biological functions of interest (Suppl. Table 1) was also analyzed using the GSEA tool[74]. Normalized read counts for Ensembl genes from HTSeq were used and enrichment was calculated using 1000 Gene Set permutations. Unsupervised clustering analysis was done using the web tool ClustVis (https://biit.cs.ut.ee/clustvis/).

**Consensus motif analysis**. Motif scanning was performed using the AME tool from the MEME Suite using the JASPAR CORE 2016 database[43].

**Chromatin immunoprecipitation (ChIP)**. ChIPs were performed on $1–20 \times 10^6$ sorted cells. The cells were cross-linked with 1.5 mM EGS for 20 min and 1% formaldehyde for 8 min before quenching with 125 mM glycine. Cells were lysed in lysis buffer and sonicated using a Covaris E220 to generate 200–600 bp fragments[75]. Samples were immuno-precipitated with 2–5 µg of either anti- GFI1 (AF3540, R&D systems), anti-H3K4me1 (ab8895, Abcam), anti-H3K4me2 (ab11946; Abcam), anti-H3K4me3 (ab8580; Abcam) or anti-H3K9me3 (ab8898; Abcam). For antibodies see Suppl. Table 4. Libraries were generated according to Illumina's instructions. Libraries were sequenced on the Illumina Hi-seq 2000 following the manufacturer's protocols to obtain 50 bp paired-end reads. External datasets were obtained in the form of.bed files of peaks and.wig visualization tracks, aligned to the mm9 build, except for LSD1, which only included the.bed peak file. There is one replicate per sample for ChIP-Seq.

**Annotation databases used**. For gene promoters, we used the Ensembl Genes 92 database, dataset GRCh38.p12. (https://useast.ensembl.org/index.html) For enhancer regions, we used the Fantom5 human_permissive_enhancers_phase_1_and_2 enhancers (February 2015) dataset (http://fantom.gsc.riken.jp).

**Statistics and reproducibility**. Two-tailed student's $t$ test was used to calculate $p$-values where indicated and values of $p < 0.05$ were considered statistically significant. Statistical analysis was done with Graph-Pad Prism software (GraphPad Software, La Jolla, CA, USA). The number of samples is indicated in the respective Figure legends. Sample sizes were $n = 3$ in Suppl. Fig. 2b; $n = 1$ in Suppl. Fig. 2c and $n = 4$ in Suppl. Fig. 3a, b, $n = 1$ in Suppl. Figs. 6 and 7 and Figs. 3G, 4E, F. Statistical comparisons were performed using the stats::t.test R function (Fig. 6), metagene plots were produced using the metagene2 R package. On regions overlapping promoters (based on the TxDb.Mmusculus.UCSC.mm10.knownGene package) and enhancers (based on the Fantom5 database).

## Data availability

The mass spectrometry proteomics data have been deposited to the ProteomeXchange Consortium via the PRIDE[76] partner repository with the dataset identifier PXD028945 and 10.6019/PXD028945. The raw ChIP-seq and RNA-seq data have been uploaded to the GEO Datasets repository[77] (https://www.ncbi.nlm.nih.gov/gds) and are available under the following accession number: GSE173533. Previously published ChIP-seq and ATAC-Seq data, which are presented in Fig. 4 and suppl. Figure 4, are available under the following accession numbers: H3K27ac (GSM1441273), H3K27me3 (DRR023959), H3K9me3 (DRR023962), H3K4me1 (GSM1441289), and ATAC-Seq (DRR023962).

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

## Acknowledgements

We are indebted to Mathieu Lapointe for technical assistance, Marie-Claude Lavallée and Jo-Anny Bisson for excellent animal care, Eric Massicotte and Julie Lord for FACS, and cell sorting. We thank Genome Quebec for performing HT sequencing. Tarik Möröy was supported by Canada Research Chair (Tier 1) and grants from the CIHR (MOP-84238, MOP-94846, FDN-148372) and the Cancer Research Society. Charles Vadnais was supported by fellowships from the Fonds de recherche Quebec—Santé (FRQS) and from the CIHR. Halil Bagci was supported by a doctoral training award from FRQS (#33603) and NSERC Discovery Grant (RGPIN-2016-04808 to Jean-François Coté). Jean-François Coté holds the TRANSAT chair in breast cancer research. Christian Trahan was supported by a CIHR Project grant (PJT-153313 to Marlene Oeffinger).

## Author contributions

A.H.: Concept and design, collection of data, data analysis, paper writing; J.H.: Collection of data, data analysis, paper writing; C.T.: Collection of data, data analysis, paper writing; C.J.B: Analysis of genomic data, P.S: setting up BioID experiment, concept, and design, collection of data, data analysis; R.C.: protein data and western blot analyses; M.A.: setting up BioID experiment, collection of data, data analysis, K.A.: Preparation of samples, collection of data, data analysis; H.B.: Proteomic data analysis, paper writing, J.F.C.: Concept and paper writing, reagents; A.D.: Concept and analysis of genomic data; Marlene Oeffinger: Concept and paper writing, T.M.: Concept and design, data analysis, and supervision, paper writing, final approval, provision of funds.

## Competing interests

The authors declare no competing interests. The work described here was conducted in compliance with Nature Springer's ethics and biosecurity guidelines.
