## [Transparent Peer Review File · Communications Biology]

Reviewers' comments:

Reviewer #1 (Remarks to the Author):

Based on earlier and in this paper performed proximity labeling studies, Helness et al show that GFI1 is in the vicinity of various nuclear complexes including the NuRD complex following its expression in 293T cells. An interaction between GFI1 and NuRD core complex member CHD4 was confirmed in a co-ip in a human leukemia cell line. They then move on to show that GFI1 and CHD4 bind DNA independent from each other but also together and that GFI1 ablation results in loss of CHD4 binding in 2/3 of co-bound genes. This does not result in consistent changes in gene expression. By performing motif analyses following GFI1B/CHD4 ChIP-seq on GMPs the authors show a strong enrichment of IRF binding sites at regions where GFI1B binds without CHD4 co-binding. At regions where GFI1 and CHD4 both bind an enrichment of PU.1 binding motifs is observed and different binding patterns (GFI1 and CHD4 alone, co-binding) also predicted co-binding with other transcription factors based on publicly available chip-seq data. The different binding possibilities associate with subtle differences in intensities of epigenetic marks and representation of genes implicated in specific biological processes upon neutrophil development. Finally, the authors show that differences in binding patterns in maturing neutrophils associate with differences in epigenetic marks and chromatin accessibility.

Main comments

The authors perform a large series of in general well executed experiments to study the transcription factor GFI1. My major concern is that this study is rather descriptive/correlative in nature and that it lacks profound novelty. The "interaction" between GFI1B and various nuclear complexes (incl NuRD) was shown by the authors earlier as they correctly state. In proximity labeling experiments any chromatin interacting protein near GFI1 will be biotinylated. Further, GFI1 and CHD4 bind DNA separately or together and this associates with some differences in epigenetic marks but this does not result in significant consistent changes in gene expression upon GFI1 ablation. I consider these findings interesting but the biological relevance remains largely unclear.

Other comments

Throughout the ms the authors suggest direct interactions between NuRD complex components and GFI1. I feel that this is too much speculation. If the authors wish to claim direct interactions with NuRD complex members they should provide evidence for that. A caveat of this study is that all proximity labeling experiments have been performed in HEK293T cells that to the best of my knowledge do not express GFI1. Whether the observed interactions occur in maturing neutrophils remains to be seen.

For me it is unclear to which extent the other interacting complexes GFI1 may be part of function and how this would influence GFI1s function. Fig 2E shows that GFI1 mainly resides in a 2 MDa complex in leukemia cells, while the NuRD members reside in lower mol weight complexes that poorly overlap with GFI1. Because LSD1 is an important subject of this study, it is a pity Fig 2E lacks an LSD1 staining. In this blot the antibody used recognizes multiple bands in the 0.5 Md range but these seem not to be present in the 2MD. Which band(s) represent GFI1?

There is debate in literature whether LSD1 is part of the NuRD complex. In a few studies LSD1 could be co-IPed with NuRD subunits, while other have failed to do so.

Figure 3a: 2K genes are bound by GFI1, 1K by both, 3K by CHD4 in GMPs. In Figure 4a quite a different distribution is observed. What is the explanation of the authors for this difference?

In the introduction the authors refer to work that claims that LSD1 rather operates as a bridging factor instead of a demethylase. This is based on a catalytically inactive LSD1 mutant but a recent studies showed that this mutant is still catalytically active (Kim et al, Mol Cell, 2020)

The authors state "anti Co-REST antibodies precipitated both GFI1 and LSD1, but to a much lesser extent MTA2 regardless of whether GFI1 was expressed or not (Fig. 2F). This suggests that GFI1 may interact with components of the NuRD complex such as MTA2 independently from the LSD1/CoREST complex. I miss a control experiment here showing that MTA2 is not a-specifically captured. The IgG seems to capture MTA2 in Fig 2B. Have the authors considered to repeat the IPs following the use of LSD1 inhibitors?"

The differences in epigenetic marks for regions bound by Chd4 and or gfi1 in Figs 4B and D are for some very subtle (eg h3k27ac and k4me3) while the authors for example state there is an enrichment in regions bound by Gfi1 for active marks and based on this conclude that these are more active regions.

In Fig 4B and D upper middle panel there seems to be quite some sites bound by GFI1 that are supposed to only be bound by CHD4 (see black line and compare this to the expected flat blue line in the upper left panels). Why is this the case? Are the cut offs for only bound by Chd4 optimally defined? This could be relevant in light of the very subtle differences observed for chromatin modifications between the different conditions.

What is the explanation of the authors that promoters bound by gfi1 only are more accessible while the opposite is seen on enhancers (compare upper right panels of figs 4b and d).

The authors mention "...RBBP7 or GATAD2B do not seem to require the SNAG domain for interaction with GFI1 (Fig. 2D), suggesting that their binding to GFI1 may be indirect via other members of the NuRD complex." Can the authors explain in more detail how this would work?

Fig 2F: Can the authors be consistent in gene names (CoREST or RCOR1)?

Blot in Fig 2A indicates rbbp4/6. Is this because the antibody recognizes both 4 and 6?

Please double check for grammatical mistakes/typo's.

Materials and methods: Prior to SEC, 200ul of NE were cleared... Can the authors clarify the 200 ul?

Pay attention to capitals for human vs mouse genes throughout the manuscript.

Resolution of some panels in figures should be improved

Results sections: "This situation is exemplified by the promoter regions of the genes encoding the GFI1 targets Csf1 (Fig. 4C) and Csf1r (EV Fig. 4), which are both expressed in GMPs (Fig. 3E, F): Csf1 is not shown in 3E or F.

In addition, Ets2, Stat3 and Sox3 binding motifs were only present at sites occupied by GFI1 and Klf, Spi1, Runx and Atf3 binding motifs were only present at sites occupied by GFI1/CEBP α (Fig. 5B). I feel "only present" is a too strong statement here. Perhaps the authors could change this into that there was no significant enrichment for the relevant combinations.

Have others found the strong association of the IRF binding motif in GFI1 ChIP-seq experiments?

Reviewer #2 (Remarks to the Author):

Helness et al. take a combined proteomic and genomic approach in characterizing in greater detail the interaction of GFI1 with the NuRD chromatin remodelling and histone deacetylase complex. Using FLAG-tag affinity purification and BioID, the authors provide clear evidence of GFI1 interactions with NuRD components. By carrying out deletion analysis, they suggest that these interactions are

mediated primarily by the N-terminal SNAG and the C-terminal zinc finger domains of GFI1. Using size fractionation analysis, the authors also suggest that GFI1 interactions with NuRD may be independent to known GFI1B interactions with LSD1 and CoREST. The authors then employ ChIPseq and expression profiling to examine the functional consequences of GFI1 and CHD4 (a key member of NuRD) co-occupancies in murine GMPs; however, a clear correlation of GFI1B and CHD4 co-occupancy on expression profiles was not evident. Interestingly, by examining chromatin accessibility and epigenetic modifications, the authors suggest the GFI1B and CHD4 co-occupancies may associate with bivalent chromatin domains. The authors then compare GFI1 and CHD4 co-occupancies to chromatin accessibility and epigenetic modifications in GMPs versus neutrophil differentiation, to show a notable decrease in chromatin accessibility H3K4me2 marks during neutrophil differentiation compared to GMPs. Although not always easy to follow, the take-home message from the genomic work is that GFI1 and CHD4 co-occupied genes in GMPs are mostly associated with bivalent chromatin modifications which may represent a poised state of gene activation and may run counterintuitively to the established roles of GFI1B and CHD4/NuRD as repressors. Neutrophil differentiation results in an overall drop in chromatin accessibility and in active epigenetic marks in the GFI1 and CHD4 target genes, in line with the known repressive roles of the two proteins.

The work is well executed and overall well presented. The conclusions drawn by the authors from the proteomic analysis may not necessarily be supported by the evidence presented (see below) and should be re-considered, but do not necessarily detract from the basic observation of GFI1B interacting with repressive chromatin modifying/epigenetic complexes. The genomic analysis is not always clear-cut but does suggest an interesting role for GFI1B/CHD4 in setting up poised chromatin domains in early stages of hematopoiesis, reverting to repressive roles with neutrophil differentiation. The authors provide an interesting discussion which proposes, for example, different actions for these proteins in enhancers versus promoters. The discussion could be further strengthened by the authors discussing how to experimentally distinguish the different hypotheses they propose.

Overall, this manuscript should be of interest to the field of transcriptional regulation in myelopoiesis. Specific comments and corrections:

- General comment: Materials and Methods should be supplemented with descriptions of BioID experiments, nuclear extraction methods, immunoprecipitation experiments.
- In the legend of Figure 2a it should be stated that lanes labelled as "-" correspond to non-transfected cells.
- Figure 2C: the authors should comment on the fact that the CHD4 IP does not immunoprecipitate construct Gfi1N152-258-Flag.
- Figure 2C: the green box and blue ovals should be labelled in the figure as the SNAG and zinc finger domains, respectively.
- Figure 2E: the MTA2 and CoREST antibodies are shown as detecting two bands, whereas in Figures 2A, B and F they detect a single band. Since the doublets detected in Fig. 2E show differential fractionation patterns, it is important that the authors indicate which of the two bands detected in Fig. 2E is real.
- The claim that "GFI1 associates with large complexes such as the NuRD and CoREST" is not really supported by the fractionation profiles shown in Fig. 2F as the GFI1 protein elutes with a peak at 2MDa which does not coincide with the elution peaks of the NuRD members CHD4 and MTA2, or of CoREST. This begs the question as to whether there is another, as yet unknown, major protein complex co-fractionating with the peak of GFI1B in the ~2MDa fractions? Also, the protein bands detected by the GFI1 antibody in fractions 13-18 vary between three bands (fractions 13-14), two bands (fractions 16-17) and a single band (fractions 17-18) which differ in mobilities between them and in their fractionation patterns. Are (any of) these bands detected truly corresponding to GFI1B protein?
- The elution profiles of MTA2 and CoREST in Figure 2E that are almost identical (also in the higher molecular weight fractions) and the co-IP results shown in Fig. 2F, do not support the claim that GFI1 interacts with MTA2 independently of CoREST. In figure 2F in particular, it is essential that IgG controls of the MTA2 and CoREST IPs are shown, in order to establish whether co-IPed proteins are enriched or not.
- Figure 3E-F and Suppl. Figure 2A: it would be good to experimentally confirm expression levels of

genes shown in these Figures by qRT-PCR.

- Suppl. Figure 3: the ChIPseq peak plot for Pbx1 is missing.
- Suppl. Figure 2B, 3A, 3B, 6 and 7 lack statistical analysis (essential for Suppl. Fig. 2B).
- Page 9, last sentence does not make sense; please revise for clarity.
- Page 8, line 15: correct Fig. 3B-D to 3C-D.
- Page 10, line 16: correct Fig. 5C to 5B.
- Legend for Suppl. Figure 1: Correct C) -D) to C) -E).
- Legend for Suppl. Figure 5: swap descriptions for panels B and C.
- The whole Western blot filters for the SEC fractionation detection of CHD4, MTA2 and CoREST are missing.

Reviewer #3 (Remarks to the Author):

Review

GFI1 tethers the NuRD complex to open and transcriptionally active chromatin in myeloid progenitors

COMMSBIO-21-1404-T

Summary:

The authors aim with this study to identify a novel physical protein interaction between GFI1 - Growth factor independent 1 transcriptional repressor and CHD4 - Chromodomain helicase DNA binding protein 4 and the resulting genomic implication. Initially, protein-protein interaction screens using GFI1 and the homologue GFI1B were performed to identify potential interaction candidates. Both screens identify CHD4, MTA2, HDAC1, Rbb4/6, LSD1 as GFI1 partners. All these proteins are part of the NuRD/CoRest complex. Next, the authors validated these interactions with targeted IP for GFI1 and GFI1B. Within the GFI1 protein the SNAG domain was identified to mediate the protein interaction. By nuclear extract separation CHD4 overlapped with the size of GFI1 suggesting presence in similar complexes. In granulocyte progenitor cells (GMP) an overlap of DNA binding site between GFI1 and CHD4 was shown. Importantly, the GFI1/CHD4 complex occupies actively transcribed genomic regions, which was shown by ATAC-seq and Chip-seq of histone methylated DNA regions. These data were used to identify DNA motifs used by GFI1, GFI1/CHD4, GFI1/CEBPa and GFI1/CEBPa/CHD4. Finally, the active chromatin status was assessed for different stages of neutrophil development (GMP, premature neutrophil, mature neutrophile) and associated cellular pathways. Overall, this study provides novel insights into the molecular/genomic network operated by GFI1 for neutrophil differentiation programs.

Major comments:

The study shows convincingly the protein interaction of GFI1 with CHD4. Moreover, the genomic architecture of GFI1 and GFI1/CHD4 was assessed properly with high-throughput methods. The manuscript is well written; however, so typo/detail editing is required.

This main limitation of the study is that no further functional evidence of CHD4 and the interaction between GFI1/CHD4 is provided for myeloid differentiation processes. Is there a specific mechanism observable that controls NuRD complex/neutrophile differentiation? Which specific steps of myeloid or lymphoid differentiation are controlled by CHD4 and how? Is there a specific add on the GFI1 function? In the reviewer's opinion the shown screens are well done; however, specific targets were not followed neither on transcriptional nor protein level. This limits the overall significance of the work. Here, the reviewer suggests further experimental evidence.

What about in vivo evidence of the CHD4/GFI1 complex? Does CHD4 deficiency leads to neutropenia

or another hematological phenotype as well? Is there a chance to model CHD4 deficiency in a simple way for validation purposes?

The authors should highlight functional evidence (expression, activation pattern of selected partners) of NuRD complex function thru neutrophil differentiation.

What about GFI1B and CHD4 in terms of functional evidence? The authors clearly show that GFI1 and GFI1B target CHD4 (and other partners) similarly. In vivo there is a relative strict, differential impact of both. Unfortunately, the authors do not follow that line.

Detailed comments:

In the Abstract the term GFI1 and CHD4 should be described fully. Both gene names are self-explaining. GFI1 - Growth factor independent 1 transcriptional repressor; CHD4 - Chromodomain helicase DNA binding protein 4

Figure legends. The figure legend should be improved as they are difficult to understand and technically overloaded. Please rephrase and clarify; avoid complex sentences. Move technical aspect that are no relevant to Mat&Meth. Clarify legends that reader instantly understands the figures. e.g. Figure legend 1 – Many abbreviations are not explained! E.g. explain AP-MS; GFI1- or GFI1B-Flag; TSS consistent label of abbreviation in figure and text e.g. Co-Rest complexes; explain BFDR

Use consistent nomenclature for CoREST, HEK293T ... !!

Please use through the manuscript (text, figures, legends) correct nomenclature for mouse Gfi1 and human GFI1.

Check several citations for correct space characters.

Check text for correct space characters.

Why did the authors perform SEC in Kasumi cells?

Please clarify GFI1 expression and the used mouse cell model. Is human GFI1 expressed in these mice? If not use for mouse genes/proteins the correct nomenclature. "We chose primary murine GMPs, in which GFI1 is expressed(Zeng et al, 2004) to further investigate the association between NuRD and GFI1." Which specific type of GMP were used.

Figure 3B, why is the diverse DNA binding of GFI1 (mainly promoter, as expected) and CHD4 (mainly intronic & distal intergenic)?

Clarify in Figure 3D, right side legend. N=841 is the overlap of binding sites between CHD4 and GFI1 in Gfi1wt. N=287 is the overlap of binding sites between CHD4 and GFI1 in Gfi1-/- . In the legend this is not clear.

Clarify in Figure 6A, 6B image that A is for promoter regions and B is for enhancer elements.

Clarify in Figure 7C specific targets that are included for the very common pathways.

Please highlight known GFI1 interaction partners in Figure 1 and 2.

Response to reviewer comment on MS COMMSBIO-21-1404-T (Helness et al.)

Reviewer #1:

Throughout the ms the authors suggest direct interactions between NuRD complex components and GFI1. I feel that this is too much speculation. If the authors wish to claim direct interactions with NuRD complex members they should provide evidence for that.

We have changed the wording to state the components associate or enter into an association rather than directly interact (see revised version of the MS, text passages marked in red on **page 2, page 6, line 13, page 7, lines 3, 10, 20, 31**).

A caveat of this study is that all proximity labeling experiments have been performed in HEK293T cells that to the best of my knowledge do not express GFI1. Whether the observed interactions occur in maturing neutrophils remains to be seen.

The BioID experiments were done in HEK 293T cells since the technology in the lab and at our institution was established in these cells and expression of transfected vectors is readily achieved. We show nevertheless that the association between NuRD complex members and GFI1 occurs at endogenous expression levels in myeloid cells such as Kasumi and THP-1 cells (See Fig. 2). In addition, we show co-occupation of GFI1 and CHD4 in ChIP-Seq experiments in myeloid progenitors i.e. GMPs. We feel that with these independent lines of evidence support well that components of the NuRD complex and GFI1 interact.

For me it is unclear to which extent the other interacting complexes GFI1 may be part of function and how this would influence GFI1s function. Fig 2E shows that GFI1 mainly resides in a 2 MDa complex in leukemia cells, while the NuRD members reside in lower mol weight complexes that poorly overlap with GFI1. Because LSD1 is an important subject of this study, it is a pity Fig 2E lacks an LSD1 staining. In this blot the antibody used recognizes multiple bands in the 0.5 Md range but these seem not to be present in the 2MD. Which band(s) represent GFI1?

We have re-probed the Western blot in Fig. 2E with an anti LSD1 antibody and can detect a band for LSD1 (see revised Fig. 2) and revised text on **page 7, last paragraph**. It is likely that the multiple GFI1 bands are a consequence of post translational modifications. We have indicated the GFI1 band according to input size by an asterisk in the revised version of the Figure 2E.

There is debate in literature whether LSD1 is part of the NuRD complex. In a few studies LSD1 could be co-IPed with NuRD subunits, while other have failed to do so.

We agree and our introduction now acknowledges this situation at **page 3, last line**.

Figure 3a: 2K genes are bound by GFI1, 1K by both, 3K by CHD4 in GMPs. In Figure 4a quite a different distribution is observed. What is the explanation of the authors for this difference?

Fig. 4 A shows only sites around TSS (i.e. transcription start sites), whereas Fig. 3A takes all peaks into account, which explains the difference.

In the introduction the authors refer to work that claims that LSD1 rather operates as a bridging factor instead of a demethylase. This is based on a catalytically inactive LSD1 mutant but a recent studies showed that this mutant is still catalytically active (Kim et al, Mol Cell, 2020)

We thank the reviewer for pointing this out. We have added a sentence to this effect into the introduction on **page 3, end of first paragraph** and are citing this reference now (**Ref 14**).

The authors state “ anti Co-REST antibodies precipitated both GFI1 and LSD1, but to a much lesser extent MTA2 regardless of whether GFI1 was expressed or not (Fig. 2F). This suggests that GFI1 may interact with components of the NuRD complex such as MTA2 independently from the LSD1/CoREST complex. I miss a control experiment here showing that MTA2 is not a-specifically captured. The IgG seems to capture MTA2 in Fig 2B. Have the authors considered to repeat the IPs following the use of LSD1 inhibitors?

We have performed a new and better controlled experiment with HEK293T cells transfected with GFI1-Flag fusion proteins. Precipitation with anti-Flag agarose beads but not beads alone show a specific interaction between GFI1 and MTA2, GFI1 and RCOR1 and GFI1 and LSD1. The input control is now also shown for both transfected and non-transfected cells (see suppl. Fig. 1F). Concerning the reviewer’s question whether we considered using an LSD1 inhibitor: this is indeed an interesting experiment; we feel however that this is rather outside the scope of this paper as LSD1 isn’t under study here.

The differences in epigenetic marks for regions bound by Chd4 and or gfi1 in Figs 4B and D are for some very subtle (eg h3k27ac and k4me3) while the authors for example state there is an enrichment in regions bound by Gfi1 for active marks and based on this conclude that these are more active regions. In Fig 4B and D upper middle panel there seems to be quite some sites bound by GFI1 that are supposed to only be bound by CHD4 (see black line and compare this to the expected flat blue line in the upper left panels). Why is this the case? Are the cut offs for only bound by Chd4 optimally defined? This could be relevant in light of the very subtle differences observed for chromatin modifications between the different conditions.

The differences in epigenetic marks for regions bound by CHD4 and or GFI1 in Figs 4B and D are due to background signal since in some cases reads are counted which do not necessarily represent sites. This appears because we wished to keep cut off parameters identical throughout the experiment. More specifically, the signal in Fig. 4B and D represent the intensity of the reads that were mapped in the selected regions. It is possible for a genomic region to contain reads and still not be considered as a valid peak during the peak calling process (i.e.: if the enrichment compared to the control is not strong enough or if the distribution of the reads is not the one that is expected for a valid peak). To make sure all the regions are comparable between samples, we also use the same parameters for the peak calling step in the same category (narrow or broad).

What is the explanation of the authors that promoters bound by gfi1 only are more accessible while the opposite is seen on enhancers (compare upper right panels of figs 4b and d).

The blue line in Fig. 4B (upper right, ATAC-seq) has a higher peak at the TSS than the blue line in Fig. 4D (ATAC-seq) i.e. at the center of the enhancer, where the peak is lower. Hence one could argue that the region at GFI1 occupied promoters is more accessible. However, we agree with the reviewer that the pattern of the ATAC-seq is also different at sites 5’ and 3’ of enhancers than at promoters when GFI1 is present. This is also seen in the Metagene analysis in Fig. 6 and to take this into account, we have added the following statement into the discussion (**page 16, lines 17-23**): “Aggregation plots from ATAC-seq analyses suggest more accessibility at promoter sites at a narrow region around the TSS compared to the enhancer sites. However, the pattern of the ATAC-seq plot is different at enhancers where it is broader compared to promoters, where it is narrower. This does not allow to conclude that promoters are more accessible than enhancers but rather indicates that different mechanisms are probably in place controlling accessibility at enhancer versus promoter sites occupied by GFI1”.

The authors mention "...RBBP7 or GATAD2B do not seem to require the SNAG domain for interaction with GFI1 (Fig. 2D), suggesting that their binding to GFI1 may be indirect via other members of the NuRD complex." Can the authors explain in more detail how this would work?

The reviewer is correct, and we apologize for this confusing statement. We have changed the sentence and write now that "In contrast, association of RBBP7 and/or GATAD2B with GFI1 is still detectable with the Δ SNAG mutant indicating that they could associate with another region of GFI1 such as the ZF domain". (See **page 7, line 20-22**)

Fig 2F: Can the authors be consistent in gene names (CoREST or RCOR1)?

We apologize for this and will stay with the name RCOR1 throughout when we address the protein, but keep the name CoREST complex which contains RCOR1, -2 and -3. We hope that this is acceptable.

Blot in Fig 2A indicates rbbp4/6. Is this because the antibody recognizes both 4 and 6?

The antibody used was from Cell Signaling (#4633S) and recognizes both RBAP46 and RBAP48; also known as RBBP7 and RBBP4. We apologize for this error and thank the reviewer for pointing this out. We have corrected this in **Fig. 2A**.

Please double check for grammatical mistakes/typo's.

We have checked the manuscript again for typos and grammatical mistakes to the best of our abilities.

Materials and methods: Prior to SEC, 200ul of NE were cleared... Can the authors clarify the 200 ul?

This has been corrected to 200 μ l

Pay attention to capitals for human vs mouse genes throughout the manuscript.

We have made the appropriate corrections in the text and figures as well as suppl. Figures.

Resolution of some panels in figures should be improved

This unfortunately was the best resolution unless the resolution is adjusted on a scanned image, though we tried to avoid this. We have replaced the pictures for the GO analysis in revised Fig. 7 for better quality.

Results sections: "This situation is exemplified by the promoter regions of the genes encoding the GFI1 targets Csf1 (Fig. 4C) and Csf1r (EV Fig. 4), which are both expressed in GMPs (Fig. 3E, F): Csf1 is not shown in 3E or F.

Csf1 expression is indeed in suppl. Fig 2A, we have corrected this mistake in the text and thank the reviewer for pointing this out.

In addition, Ets2, Stat3 and Sox3 binding motifs were only present at sites occupied by GFI1 and Klf, Spi1, Runx and Atf3 binding motifs were only present at sites occupied by GFI1/CEBP α (Fig. 5B). I feel "only present" is a too strong statement here. Perhaps the authors could change this into that there was no significant enrichment for the relevant combinations.

We have changed the sentence and have replaced "only present" by highly enriched". The sentence reads now: "In addition, ETS2, STAT3 and SOX3 binding motifs were highly enriched at sites occupied by GFI1, but in contrast, KLF, SPI1, RUNX and ATF3 binding motifs were highly enriched at sites occupied by GFI1/CEBP α "; see revised text on **page 10, line 14-16**.

Have others found the strong association of the IRF binding motif in GFI1 ChIP-seq experiments?

Yes, CHIP-Seq analysis of GFI1 and IRF8 in GMPs showed statistically enriched GFI1 ($p=6.63 \times 10^{-8}$) and ETS-IRF composite element (EICE) motifs ($p=1.08 \times 10^{-6}$) in Olsson et al., Nature 2016 (PMID: 27580035). We had already mentioned this in the Discussion (See **page 17, line 5-7**).

Reviewer #2:

- General comment: Materials and Methods should be supplemented with descriptions of BioID experiments, nuclear extraction methods, immunoprecipitation experiments.

We have now included a detailed description of the BioID experiment, for the nuclear extraction method and the SEC (see Materials and Methods section, **pages 18, 19 and 20**). The immunoprecipitation methods have been described before in our publications Shooshtarizadeh et al., Nat Comm 2019 and Vadnais et al., Nat Comm. 2018.

- In the legend of Figure 2a it should be stated that lanes labelled as “-“ correspond to non-transfected cells.

We have added the following sentence to the legend of Fig. 2A and 2F: Lanes with extracts from cells transfected or not with constructs for the expression of GFI1-Flag fusion proteins are labelled as “+” or “-“, respectively (see **page 27, line 3-5**)

- Figure 2C: the authors should comment on the fact that the CHD4 IP does not immunoprecipitate construct Gfi1N152-258-Flag.

We mention this now in the Results section on **page 7, line 12-14**.

- Figure 2C: the green box and blue ovals should be labelled in the figure as the SNAG and zinc finger domains, respectively.

We have labeled both domains as requested in **Fig. 2C**

- Figure 2E: the MTA2 and CoREST antibodies are shown as detecting two bands, whereas in Figures 2A, B and F they detect a single band. Since the doublets detected in Fig. 2E show differential fractionation patterns, it is important that the authors indicate which of the two bands detected in Fig. 2E is real.

We have indicated the GFI1 band with an asterisk in Fig. 2E. For MTA2 and RCOR1, we have done a new immunoprecipitation and western blot experiment with extracts from GFI1-Flag transfected HEK293 cells (see **suppl. Fig. 1F**). We have obtained a better resolution of the bands for MTA2 and RCOR1 indicating that GFI1 associates with a protein represented by the lower MTA2 band and the higher RCOR1 band (see **suppl. Fig. 1F**). We have indicated these bands in the SEC elution western blot (**Fig. 2E**) with an asterisk and explain this in the Figure legend (see revised MS, **legend to Fig. 2E, page 27**).

- The claim that “GFI1 associates with large complexes such as the NuRD and CoREST” is not really supported by the fractionation profiles shown in Fig. 2F as the GFI1 protein elutes with a peak at 2MDa which does not coincide with the elution peaks of the NuRD members CHD4 and MTA2, or of CoREST. This begs the question as to whether there is another, as yet unknown, major protein complex co-fractionating with the peak of GFI1B in the ~2MDa fractions?

This is likely the case. We have re-probed the blot with an LSD1 antibody and can detect LSD1 in fractions 13 and 14; i.e. in the 0.5 MDa complex, but not in the 2 MDa complex (See **revised Fig. 2E**). This supports our statement the GFI1 is part of the NuRD complex together with LSD1 and RCOR1 and that this complex elutes at around 0.5M MDa. Since LSD1 is almost absent from the 2 MDa complex (fractions 2 and 3) where a substantial amount of GFI1 is found, it is likely that another, unknown GFI1 complex exists that does not contain LSD1 nor RCOR1 nor members of the NuRD complex. The nature of this complex remains to be determined in future studies.

- Also, the protein bands detected by the GFI1 antibody in fractions 13-18 vary between three bands (fractions 13-14), two bands (fractions 16-17) and a single band (fractions 17-18) which differ in

mobilities between them and in their fractionation patterns. Are (any of) these bands detected truly corresponding to GFI1B protein?

Our new immunoprecipitation and western blot experiment with extracts from GFI1-Flag transfected HEK293T cells indicates that GFI1 associates with MTA2, besides LSD1 and RCOR1 (see new suppl. Fig. 1F). We are sure that the band to which we have added an asterisk in the image of Fig. 2E represents GFI1 according to the input and data in suppl. Fig. 1F

- The elution profiles of MTA2 and CoREST in Figure 2E that are almost identical (also in the higher molecular weight fractions) and the co-IP results shown in Fig. 2F, do not support the claim that GFI1 interacts with MTA2 independently of CoREST. In figure 2F in particular, it is essential that IgG controls of the MTA2 and CoREST IPs are shown, in order to establish whether co-IPed proteins are enriched or not.

We have performed a new experiment with HEK293T cells transfected with GFI1-Flag fusion proteins. Precipitation with anti-Flag agarose beads and beads alone show a specific interaction between GFI1 and MTA2, GFI1 and RCOR1, GFI1 and LSD1. The input control is now also shown for both transfected and non-transfected cells (see new suppl. Fig. 1F).

- Figure 3E-F and Suppl. Figure 2A: it would be good to experimentally confirm expression levels of genes shown in these Figures by qRT-PCR.

We have done Q-PCR analysis of these genes and have included the results in a new suppl. Fig. 2B

- Suppl. Figure 3: the ChIPseq peak plot for Pbx1 is missing.

We have added the tracks for Pbx1 to suppl. Fig. 3.

- Suppl. Figure 2B, 3A, 3B, 6 and 7 lack statistical analysis (essential for Suppl. Fig. 2B).

We have now added a statistical analysis for suppl. Fig. 3A and B.

The RNA-seq data in suppl. Fig. 6 and 7 were shown to confirm that the sorted GMP, preNeu and matNeu populations indeed represent these subsets. Since the RNA-Seq data follow exactly the RNA expression pattern that were previously published for these subpopulations (see reference 44, Evrard, M. *et al.* Developmental Analysis of Bone Marrow Neutrophils Reveals Populations Specialized in Expansion, Trafficking, and Effector Functions. *Immunity* 48, 364-379 e368,).

We feel that this point has been made (our sorted cells behave as the published cells) and – if the reviewers agree with this view – would therefore leave the data set as is. We do only have one RNA-seq data set of the GMP, preNeu and matNeu subsets and a statistical analysis would require two-three repetitions of RNA-Seq experiments.

The experiment in suppl. Fig. 2B (now suppl. Fig. 2C) would have to be repeated if a statistical analysis is required, since we have only one data set. A repetition of the experiment is possible, but would require to sacrifice a very large number of animals again: we would require 10 million GMPs for the anti-CHD4 and for the anti-Ig control ChIP each for both wt and Gfi1 KO mice, i.e. 40 million cells. We need bone marrow from 10 wt mice to obtain 4 million wt GMPs and 5-6 mice for 4 million GMPs from Gfi1 KO mice after FACS sorting. Also, the Gfi1 KO mouse cohort would have to be expanded, which would take a few additional months, since we do not have these large numbers of KO animals right now. If it is absolutely required to repeat this experiment for the acceptance of the paper, we will certainly engage in doing this, however, we feel that the existing data support the point that has been made to demonstrate that a fraction of genes lose CHD4 occupation when GFI1 is absent with the data (ChIP-Seq and ChIP-PCR data) as they stand.

- Page 9, last sentence does not make sense; please revise for clarity.

We apologize for not being clear. We have rewritten this part in two sentences as follows: “These findings indicate that GF11/CHD4 complexes occupy active or bivalent promoters and also active enhancers. These sites have a more closed chromatin configuration when CHD4 is present and have a more open chromatin conformation when GF11 is present” (see page 9, last 2 lines and page 10, lines 1 and)

- Page 8, line 15: correct Fig. 3B-D to 3C-D.

This has been corrected

- Page 10, line 16: correct Fig. 5C to 5B.

This has been corrected

- Legend for Suppl. Figure 1: Correct C) -D) to C) -E).

This has been changed

- Legend for Suppl. Figure 5: swap descriptions for panels B and C.

The description for panels B and C has been interchanged, we are sorry for the mistake

- The whole Western blot filters for the SEC fractionation detection of CHD4, MTA2 and CoREST are missing.

We have included the missing scans and apologize for this oversight

Reviewer #3:

Major comments:

What about in vivo evidence of the CHD4/GFI1 complex? Does CHD4 deficiency leads to neutropenia or another hematological phenotype as well? Is there a chance to model CHD4 deficiency in a simple way for validation purposes?

According to published reports, CHD4 is expressed highly in hematopoietic stem cells, early lymphoid progenitors, myeloid progenitors and erythroid progenitors (reviewed in PMID: 19923891). Deletion of CHD4 leads to expansion of hematopoietic stem cells, which then show a lymphoid bias and a loss or reduction of Mac-1+/Gr1+ granulocytes and B-cell precursors (B220+CD19+), and in increase in erythroid precursors (see PMID 18451107). It would therefore be difficult to use CHD4 KO mice to test its function in association with GFI1 in neutrophil granulocytes.

The authors should highlight functional evidence (expression, activation pattern of selected partners) of NuRD complex function thru neutrophil differentiation.

Information on the activity and expression of NuRD complex members in early myeloid differentiation is limited, but as mentioned above, CHD4 was found to be expressed in hematopoietic stem cells, early lymphoid progenitors, myeloid progenitors and erythroid progenitors (PMID 18451107). According to this paper, ablation of CHD4 in mice (in vivo) leads to a loss or reduction of reduction of Mac-1+/Gr1+ granulocytes, which are the same cells affected by GFI1 depletion. We mention this now in the Discussion section on **page 17, end of last paragraph**) and write: "It has been reported that the deletion of *Chd4* in mice leads to the reduction of neutrophil granulocytes and B-cell precursors, which are the same cells that require GFI1 and to an increase in erythroid precursors, which are known to require GFI1B (see PMID 18451107). Although the biological role of the interaction of GFI1 and GFI1B with members of the NuRD complex remains to be investigated in more detail, these findings suggest that the association of GFI1 and GFI1B with the NuRD complex is important for all early hematopoietic lineages".

However, it would be difficult to highlight other evidence on the functional role of NuRD complex in myeloid differentiation, given the sparse literature on the subject.

What about GFI1B and CHD4 in terms of functional evidence? The authors clearly show that GFI1 and GFI1B target CHD4 (and other partners) similarly. In vivo there is a relative strict, differential impact of both. Unfortunately, the authors do not follow that line.

In mice, GFI1B is restricted to the erythroid and megakaryocytic lineage, although GFI1B is also expressed in the leukemia lines we have used here. There are few data on NuRD complex members in erythroid differentiation, but one paper describes that the disruption of FOG1/NuRD affects erythroid and megakaryocytic differentiation. The authors suggest that a continuous presence of GATA1, FOG1, and NuRD complex members is required to maintain a MK-erythroid lineage (PMID: 20065294), just as is the case for GFI1B. We mention this now in the Discussion on **page 17, end of last paragraph**) and write: It has been reported that the deletion of *Chd4* in mice leads to the reduction of neutrophil granulocytes and B-cell precursors, which are the same cells that require GFI1 and to an increase in erythroid precursors, which are known to require GFI1B (see PMID 18451107). Although the biological role of the interaction of GFI1 and GFI1B with members of the NuRD complex remains to be investigated in more detail, these findings suggest that the association of GFI1 and GFI1B with the NuRD complex is important for all early hematopoietic lineages.

Since we have focused on GFI1 in our study with differentiated myeloid cells, we feel that the investigation of the role of the GFI1B/NuRD complex interaction in erythroid and megakaryocytic differentiation would require a new, independent study.

Detailed comments:

In the Abstract the term GFI1 and CHD4 should be described fully. Both gene names are self-explaining. GFI1 - Growth factor independent 1 transcriptional repressor; CHD4 - Chromodomain helicase DNA binding protein 4

We have added the full names of Growth factor independent 1 and Chromodomain helicase DNA binding protein 4 into the Abstract as requested.

Figure legends. The figure legend should be improved as they are difficult to understand and technically overloaded. Please rephrase and clarify; avoid complex sentences. Move technical aspect that are not relevant to Mat&Meth. Clarify legends that reader instantly understands the figures.

e.g. Figure legend 1 – Many abbreviations are not explained! E.g. explain AP-MS; GFI1- or GFI1B-Flag; TSS consistent label of abbreviation in figure and text e.g. Co-Rest complexes; explain BFDR

We apologize for being unclear in our Figure legend descriptions. We have now defined abbreviations such as AP-MS, which was already defined in the text (**page 6, line 2**) also in the legend to Figure 1. We write now that GFI1- and GFI1-Flag are Flag tagged fusion proteins and we have also spelled out TSS in the legends to Figures that depict aggregation plots. We have spelled out BFDR (which stands for “Bayesian False Discovery Rate”) in the legend to **Fig. 1C., Fig. 2D.**

We went through the text of the Figure legends and have added more explanations to Fig. 2. We feel, however, that removing text from the legends to Materials and Methods will make the description less clear. We hope, however, that the changes we made have rendered the Figure legends more intelligible.

Use consistent nomenclature for CoREST, HEK293T ... !!

We have used the term RCOR now throughout the manuscript but keep the term “LSD1/CoREST complex”, when we mention complexes to follow what is used in GO terms. We hope that this is acceptable. We now consistently use the term HEK293T throughout the MS.

Please use through the manuscript (text, figures, legends) correct nomenclature for mouse Gfi1 and human GFI1.

We now use italics for the murine *Gfi1* gene and capital letter for the murine GFI1 protein throughout including Figures, suppl. Figures and in the text. The human GFI1 protein or *GFI1* gene is not mentioned in the text.

Check several citations for correct space characters.

We went through the citations in the text and removed spaces between the end of the sentence and the citation number.

Check text for correct space characters.

We went through the text and removed superfluous spaces.

Why did the authors perform SEC in Kasumi cells?

Kasumi cells express GFI1 and CHD4 and we had detected an interaction of GFI1 and MTA2 and GFI1 and CHD4 in Kasumi cells. These cells seemed therefore to be best suitable for this experiment.

Please clarify GFI1 expression and the used mouse cell model. Is human GFI1 expressed in these mice? If not use for mouse genes/proteins the correct nomenclature. "We chose primary murine GMPs, in which GFI1 is expressed (Zeng et al, 2004) to further investigate the association between NuRD and GFI1."

The cells used here are from wild type mice. GMPs have been shown to express GFI1 (Zeng et al., EMBO J, 2004); we show in Western blot and FACS analyses that GFI1 is expressed during neutrophilic differentiation (see **suppl. Fig. 4**). The mice used here are **not** transgenic for a human GFI1, hence express only the endogenous murine GFI1. The GMPs from GFI1 KO mice lack GFI1 (see Karsunky et al., Nat. Genetics, 2002)

Which specific type of GMP were used.

GMPs used here have been obtained through FACS sorted from mouse bone marrow. The marker phenotype (lin⁻cKIT⁺SCA1⁻CD16/32⁺CD34⁺) is indicated in the legend of **suppl. Fig. 5**. We have now added the GMP marker phenotype into the text in the results section on **page 8, line 4**.

Figure 3B, why is the diverse DNA binding of GFI1 (mainly promoter, as expected) and CHD4 (mainly intronic & distal intergenic)?

CHD4 and the NuRD complex acts as a chromatin remodeler, i.e. a machine that displaces histones and opens or closes chromatin throughout a wide variety of chromatin regions. In contrast, GFI1 is a transcription factor with a specific consensus DNA binding site within regulatory sites near genes either in promoters or enhancers and much less widely dispersed within the genome. This would explain the diverse pattern seen in Fig. 3B.

Clarify in Figure 3D, right side legend. N=841 is the overlap of binding sites between CHD4 and GFI1 in Gfi1wt. N=287 is the overlap of binding sites between CHD4 and GFI1 in Gfi1^{-/-}. In the legend this is not clear. Clarify in Figure 6A, 6B image that A is for promoter regions and B is for enhancer elements.

Clarify in Figure 7C specific targets that are included for the very common pathways.

Please highlight known GFI1 interaction partners in Figure 1 and 2.

- We have added the following sentence to the legend of Figure 3D: N=841: sites occupied by both CHD4 and GFI1 in wt GMPs which lose CHD4 occupation when *Gfi1* is deleted (i.e. in *Gfi1* KO GMPs). N=287: sites occupied by both CHD4 and GFI1 in wt GMPs which maintain CHD4 occupation when *Gfi1* is deleted (i.e. in *Gfi1* KO GMPs).
- The Figure legend 6A and B already states that A concerns promoter regions and B enhancers.
- Figure 7C: we now mention in the legend that examples of genes for the pathways are given in **suppl. Table 4**.
- We have highlighted known GFI1 interacting proteins by asterisks in **Fig. 1A, B, C** and in **Fig. 2D**.

REVIEWERS' COMMENTS:

Reviewer #1 (Remarks to the Author):

The authors have adequately addressed the issues and questions I raised.

I feel that the presented findings are interesting and that this manuscript contains many new data but still consider it correlative in nature as exemplified by the title of the manuscript.

I noticed a few minor things

P7, sentence "Neither CHD3 not MTA2.." should probably read Neither CHD3 nor MTA2...

Both KDM1A and LSD1 are being used throughout the ms, figures legends and figures. For consistency please choose one (KDM1A is the official name).

The authors give a good explanation considering my question as to the differences in epigenetic marks for regions bound by CHD4 / GFI1 in Figs 4B and D. It would help if they could add their reasoning in short to the legends of that figure.

With my question regarding 200 ul I actually meant what the protein concentration used was.

Reviewer #2 (Remarks to the Author):

The authors have addressed most of comments satisfactorily however, two issues remain:

1. the detection of GFI1 protein by the anti-GFI1 antibody in figure 2E remains somewhat ambiguous, in that the band indicated by an asterisk in lanes 17 and 18 falls between the two bands detected in lanes 15, 16 and again a different pattern is detected in fractions 13, 14. This is an important point to clarify as the authors suggest that fractions 13 and 14 represent the peak of GFI1 co-fractionation with CHD4, MTA2, LSD1 and RCOR1, but it is not clear which is the real GFI1 protein band detected by the antibody in these two fractions.

2. The response of the authors does not address my comment about co-IP and co-fractionation of MTA2 and CoREST in Fig. 2D and F.

Minor comment: the newly added methods need to be edited, e.g. remove "got about 2.5 mL each" (second sentence of the nuclear extracts methods description).

The response of the authors regarding the lack of statistical analysis in suppl. fig. 2b (2C in the revised manuscript) is acceptable.

Reviewer #3 (Remarks to the Author):

The authors responded to all major and minor comments.

The authors improved the manuscript significantly.

The major limitation of the study, validation of the in vitro findings in an appropriated in vivo model, was not addressed with further experiments. This limitation was mentioned in the text.

In the overall view I Reviewer #3 supports publication of the manuscript.

--

Response Reviewer #3 to major comment A.

The study 18451107 uses an inducible conditional Chd4 (Mi-2 β) model. Depending on the time after homozygous ko-induction the number of granulocytes declines. Whether the still existing granulocytes are already ko-cells was not characterized. The authors ignore the possibility to test the heterozygous

Chd4 (Mi-2 β) model.

The reviewer agrees that in vivo characterization of the Cdh4 and Gfi1 interaction is challenging and may exceed the scope of the current study. However, this evidence is critical to prove the hypothesis generated by the authors.

Response Reviewer #3 to major comment B.

The reviewer agrees that in vivo characterization of the Cdh4 and Gfi1 interaction is challenging and may exceed the scope of the current study. However, this evidence is critical to prove the hypothesis generated by the authors. The authors should think about using an in vivo approach e.g. heterozygous cells to solicit their hypothesis.

Response Reviewer #3 to major comment C.

The study 18451107 provides strong evidence that deactivation of Chd4 has a broad impact on hematopoietic lineage development. This should motivate further in vivo analysis of Gfi1 and Gfi1b as NuRD/Chd4 interaction partner.

Response Reviewer #3.

Figure 2 has in figure legend "D" twice.

Response Reviewer #3. The correct nomenclature is Gfi1 gene and Gfi1 protein. Full capitalization is used only for human GFI1 and GFI1 protein. This is true for all genes and proteins. Hek293 cells are human cells.

Reviewer #1 (Remarks to the Author):

P7, sentence “Neither CHD3 not MTA2.. “ should probably read Neither CHD3 nor MTA2... This was corrected

Both KDM1A and LSD1 are being used throughout the ms, figures legends and figures. For consistency please choose one (KDM1A is the official name).

We introduce LSD1 as KDM1A on page 4 and thereafter use the term LSD1. The text was revised on page 3 as follows:

The DNA binding zinc finger proteins GFI1 and GFI1B act as transcriptional repressors by recruiting the complex containing the Lysine-Specific Histone Demethylase 1A (LSD1, official name KDM1A) and its co factor CoREST (official name RCOR1) together with HDACs to sites of specific target genes that harbor a GFI/GFI1B consensus DNA binding motif

The authors give a good explanation considering my question as to the differences in epigenetic marks for regions bound by CHD4 / GFI1 in Figs 4B and D. It would help if they could add their reasoning in short to the legends of that figure.

We have added the following sentence to the legend of Fig 4B and D on page 29:

The differences in epigenetic marks for regions bound by CHD4 and or GFI1 in B) and D) (see below) are due to background signal since in some cases reads are counted which do not necessarily represent sites, because cut off parameters were kept identical throughout the experiment. The signal in B) and D) represent the intensity of the reads that were mapped in the selected regions. To ensure all regions are comparable between samples, the same parameters were used for the peak calling step in the same category.

With my question regarding 200 ul I actually meant what the protein concentration used was.

The protein concentration was not determined before loading the column.

Reviewer #2 (Remarks to the Author):

1. the detection of GFI1 protein by the anti-GFI1 antibody in figure 2E remains somewhat ambiguous, in that the band indicated by an asterisk in lanes 17 and 18 falls between the two bands detected in lanes 15, 16 and again a different pattern is detected in fractions 13, 14. This is an important point to clarify as the authors suggest that fractions 13 and 14 represent the peak of GFI1 co-fractionation with CHD4, MTA2, LSD1 and RCOR1, but it is not clear which is the real GFI1 protein band detected by the antibody in these two fractions.

We have clarified in the legend to Fig. 2E (on page 27) that the bands marked with an asterisk represent the GFI1 protein and have added the sentence: It is likely that the bands represent differentially modified forms of the GFI1 protein, since they are recognized by the anti GFI1 antibody. This is stated in the legend to Fig. 2E.

2. The response of the authors does not address my comment about co-IP and co-fractionation of MTA2 and CoREST in Fig. 2D and F.

In suppl. Fig. 1F, we show a new experiment with a better gel resolution that indicates that MTA2 is detected as two bands, but only the lower (Major band) associated with GFI1. Similarly CoREST is detected as two bands, but only the upper (major band) associates with GFI1. We have therefore put an asterisk next to the bands of MTA2 and CoREST in the gel in Fig. 2 (see input lane) to indicate which band represent the MTA and CoREST forms that associate with GFI1.

Minor comment: the newly added methods need to be edited, e.g. remove "got about 2.5 mL each" (second sentence of the nuclear extracts methods description).

This has been removed

Reviewer #3 (Remarks to the Author):

The study 18451107 uses an inducible conditional Chd4 (Mi-2^{fl}) model. Depending on the time after homozygous ko-induction the number of granulocytes declines. Whether the still existing granulocytes are already ko-cells was not characterized. The authors ignore the possibility to test the heterozygous Chd4 (Mi-2^{fl}) model.

We agree that this would be indeed an attractive model and a follow up with these mice may be . However, this new mouse model would have to be tested for its suitability for our current study. For instance, it would have to be clarified how many remaining granulocytes can be captured after KO-induction and whether these cells are amenable to experimentation. Also, it is not known whether heterozygous Chd4 (Mi-2^{fl}) mice have a reduced CHD4 expression and whether this leads to a milder phenotype in myeloid differentiation. All this is not established and would have to be tested beforehand. Since the outcome of this is not known, we feel that would take is far beyond the frame of the study

The reviewer agrees that in vivo characterization of the Cdh4 and Gfi1 interaction is challenging and may exceeds the scope of the current study. However, this evidence is critical to prove the hypothesis generated by the authors.

We agree and will try to follow up on the reviewer's suggestion to use the Chd4 Ko model for future studies.

The reviewer agrees that in vivo characterization of the Cdh4 and Gfi1 interaction is challenging and may exceeds the scope of the current study. However, this evidence is critical to prove the hypothesis generated by the authors. The authors should think about using an in vivo approach e.g. heterozygous cells to solicit their hypothesis.

We agree and will try to follow up on the reviewer's suggestion to use the Chd4 Ko model for future studies.

The study 18451107 provides strong evidence that deactivation of Chd4 has a broad impact on hematopoietic lineage development. This should motivate further in vivo analysis of Gfi1 and Gfi1b as NuRD/Chd4 interaction partner.

We agree and will try to follow up on the reviewer's suggestion to use the Chd4 KO model for future studies to investigate the role of the GFI1/NuRD interaction in hematopoiesis and myeloid differentiation in the future.

Figure 2 has in figure legend "D" twice.

This was corrected

The correct nomenclature is Gfi1 gene and Gfi1 protein. Full capitalization is used only for human GFI1 and GFI1 protein. This is true for all genes and proteins. Hek293 cells are human cells.

According to Biosciencewriters the following rules apply:

Humans: Gene symbols contain three to six italicized characters that are all in upper-case, protein symbols are identical to their corresponding gene symbols except that they are not italicized (e.g., AFP).

Mice : Gene symbols are italicized, with only the first letter in upper-case (e.g., Gfap).

Protein symbols are not italicized, and all letters are in upper-case (e.g., GFAP).